# Refine Drugs, Don't Complete Them: Uniform-Source Discrete Flows for Fragment-Based Drug Discovery

**Benno Kaech**[*]   **Luis Wyss**[†]   **Karsten Borgwardt**[†]   **Gianvito Grasso**[*]

## ABSTRACT

We introduce InVirtuoGen, a discrete flow generative model for fragmented SMILES for de novo and fragment-constrained generation, and target-property/lead optimization of small molecules. The model learns to transform a uniform source over all possible tokens into the data distribution. Unlike masked models, its training loss accounts for predictions on all sequence positions at every denoising step, shifting the generation paradigm from completion to refinement, and decoupling the number of sampling steps from the sequence length. For *de novo* generation, InVirtuoGen achieves a stronger quality-diversity pareto frontier than prior fragment-based models and competitive performance on fragment-constrained tasks. For property and lead optimization, we propose a hybrid scheme that combines a genetic algorithm with a Proximal Property Optimization fine-tuning strategy adapted to discrete flows. Our approach sets a new state-of-the-art on the Practical Molecular Optimization benchmark, measured by top-10 AUC across tasks, and yields higher docking scores in lead optimization than previous baselines. InVirtuoGen thus establishes a versatile generative foundation for drug discovery, from early hit finding to multi-objective lead optimization. We further contribute to open science by releasing pretrained checkpoints and code, making our results fully reproducible[1].

## 1 INTRODUCTION

Fragment-based drug discovery (FBDD) is widely used in both academia and industry for its efficient exploration of chemical space (Kirkman et al., 2024). FBDD relies on fragment-constrained design, in which new candidate molecules are generated by preserving specific substructures, such as active scaffolds or pharmacophores, and modifying surrounding regions to tune properties (Kirsch et al., 2019). However, FBDD is typically guided by expert-defined heuristics based on chemical intuition, limiting exploration of the vast chemical space. In contrast, *in silico* molecular generation seeks to formalize and automate this intuition, leveraging data-driven generative models to navigate chemical space more systematically. While many such models have emerged as promising candidates to accelerate the drug discovery pipeline (Zeng et al., 2022), their adoption in practice remains limited. A key barrier is that their molecular representations are often poorly aligned with medicinal chemistry workflows, making them difficult to integrate into existing pipelines. Although graphs provide a natural representation for molecules, generative frameworks tailored to graph-structured data remain limited in performance. For instance, the state-of-the-art AutoGraph (Chen et al., 2025) sidesteps direct graph generation by linearizing graphs into sequences via depth-first traversal and relying on next token prediction. Similarly, the Simplified Molecular Input Line Entry System (SMILES) (Weininger, 1988) is commonly used when working with small molecules. SMILES encodes molecular graphs as sequences via depth-first traversal of a spanning tree with annotations for branches and ring closures. However, both linearizations disrupt chemically meaningful substructures, offering limited control over scaffold retention or fragment assembly and making them poorly suited for fragment-based drug discovery (Jinsong et al., 2024). We propose InVirtuoGen, a continuous-time discrete flow model (Campbell et al., 2024; Gat et al., 2024) that operates directly on fragmented SMILES.

---

[*]In Virtuo Laboratories

[†]Max Planck Institute of Biochemistry

[1]https://github.com/invirtuolabs/InVirtuoGen_results

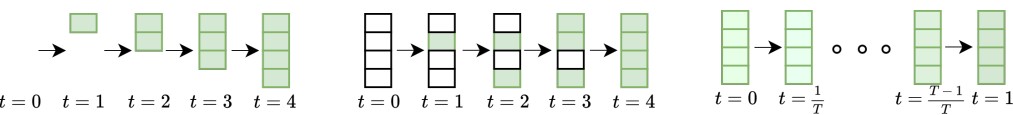

|  (a) Autoregressive | (b) Masked Diffusion | (c) Discrete Flow & Uniform Source |

Figure 1: Comparison of generation paradigms: (a) autoregressive models generate tokens sequentially (here simplified by omitting BOS/EOS tokens), (b) masked diffusion models iteratively reveal masked positions, and (c) discrete flows refine all positions starting from a uniform source distribution, where shading indicates the transition from random tokens to data.

## 2 RELATED WORK

While numerous generative models have been proposed for small-molecule drug design (Jensen, 2019; Olivecrona et al., 2017; Morrison et al., 2024; Gao et al., 2022b; Nigam et al., 2020; Bou et al., 2024), few are explicitly designed for fragment-level control. In this work, we focus on approaches that operate on sequential fragment-based representations. Several alternative approaches construct molecules via graph-based operations on substructures, for example by adding or deleting fragments through Markov Chain Monte Carlo sampling (Xie et al., 2021), using graph-based Variational Auto-Encoders conditioned on identified substructures (Jin et al., 2020; 2018; Maziarz et al., 2024), or applying graph-based genetic algorithms (GA) (Jensen, 2019; Tripp and Hernández-Lobato, 2023). Although these models encode domain-specific priors, they often suffer from poor scalability and limited generalization beyond known chemical regions, in part due to their reliance on graph operations and discrete mutation strategies. By contrast, generative models operating on linear sequential representations of fragmented SMILES, such as SAFE-GPT (Noutahi et al., 2023) and GenMol (Lee et al., 2025), offer a more scalable and expressive alternative, and form the primary baselines for our work[2].

**Autoregressive Models**  Autoregressive approaches, such as SAFE-GPT (Noutahi et al., 2023), generate molecular sequences token by token in a fixed left-to-right order. This ordering is arbitrary with respect to molecular structure, which is inherently unordered[3].

**Masked Diffusion Models**  Masked discrete diffusion models, such as GenMol (Lee et al., 2025), iteratively unmask tokens starting from a fully masked input. While predictions are produced for the entire sequence at each denoising step, the training objective accounts for errors only on the masked positions. As a consequence, once a token is unmasked during sampling, it is treated as fixed and no longer updated. This introduces a fundamental limitation: the number of sampling steps is bounded by the number of initially masked tokens, unless arbitrary remasking heuristics are included.

**Our Contributions**  Our method departs from completion-style generation and instead refines all positions simultaneously at every denoising step (Figure 1). This training paradigm enables coordinated updates across the molecule and decouples sampling steps from sequence length, aligning with our central principle: *refine drugs, don't complete them*. Concurrent to our work (Schiff et al., 2025) proposed Uniform Discrete Language Models, which similarly allow simultaneous token updates but remain within a diffusion-based framework. Concretely, we present the first discrete flow model for fragmented SMILES with a refinement-based training paradigm, show state-of-the-art performance in *de novo* generation and competitive performance on fragment-constrained generation tasks, and introduce a hybrid optimization framework combining Proximal Property Optimization (Schulman et al., 2017), adapted to discrete flows, with a genetic algorithm. Our framework achieves state-of-the-art results on the Practical Molecular Optimization (PMO) benchmark (Gao et al., 2022a) and improved results in lead optimization over prior baselines (Lee et al., 2025).

---

[2]We compare against GenMol using the results reported in their paper. While the source code is publicly available, to the best of our knowledge, the pretrained checkpoints are only distributed through NVIDIA NIM/NGC under the NVIDIA Open Model License, which currently does not allow unrestricted download. As a result, we were unable to run GenMol directly and rely on the published numbers for comparison.

[3]Although, there exist a canonical SMILES encoding scheme, it is still an arbitrary imposed ordering.

## 2.1 DISCRETE FLOW MODELS

We adopt the discrete flow model framework of Gat et al. (2024), where the goal is to transform samples from a source distribution $X_0 \sim p$ into samples from a target distribution $X_1 \sim q$. Training data consists of interpolation pairs $(X_0, X_1)$, sampled independently from the source and target. We choose the linear scheduler $\kappa_j^t$ of Gat et al. (2024), resulting in following the probability path:

$$p_t(x^i \mid x_0, x_1) = (1 - t)\, \delta_{x_0}(x^i) + t\, \delta_{x_1}(x^i), \quad t \in [0, 1]. \tag{1}$$

During sampling, each token $X_t^i$ is updated according to the discrete-time Markov update

$$X_{t+h}^i \sim \delta_{X_t^i}(\cdot) + h\, u_t^i(\cdot, X_t), \tag{2}$$

where $u_t$ is the *probability velocity* and $h > 0$ is the step size. Following Gat et al. (2024), $u_t$ must satisfy the validity constraints

$$\sum_{x^i \in [d]} u_t^i(x^i, z) = 0, \quad u_t^i(x^i, z) \geq 0 \text{ for all } i \text{ and } x^i \neq z. \tag{3}$$

For our scheduler choice, the training objective becomes

$$\mathcal{L}(\theta) = -\mathbb{E}_{t \sim U(0,1),(X_0,X_1),X_t} \frac{1}{1 - t^2} \sum_i \log p_{1|T}(X_1^i \mid X_t), \tag{4}$$

where $p_{1|T}$ denotes the model prediction and the sum is over the sequence. The time-dependent loss weighting term, not present in its original formulation, was inspired by Sahoo et al. (2024) and places greater emphasis on later timesteps, encouraging higher accuracy near the end of the trajectory. As a backbone model, we use a diffusion transformer (Peebles and Xie, 2022) to parameterize $p_{1|T}$, leveraging its bidirectional self-attention to capture long-range dependencies between fragments while predicting the target token distribution at each position. More details about our experimental setup is given in Appendix A.1.

## 2.2 FRAGMENTED SMILES NOTATION & PREPROCESSING

Our molecule representation is based on and marginally extends the Sequential Attachment-Based Fragment Embedding (SAFE) framework of Noutahi et al. (2023) by encoding molecules as sequences of fragment blocks with explicit attachment points, improving readability and direct control over molecular substructures. To produce chemically meaningful fragments, the molecules are decomposed using the revised BRICS algorithm (Degen et al., 2008), with the locations of bond breaks marked by attachment points of the form $[i*]$, where $i$ enumerates the broken bonds and we separate fragments with spaces. In Fig. 2, we illustrate the difference between the SMILES, SAFE, and our notation. To remove any implicit ordering bias, the resulting fragments are randomly shuffled rather than ordered by their attachment point in the original

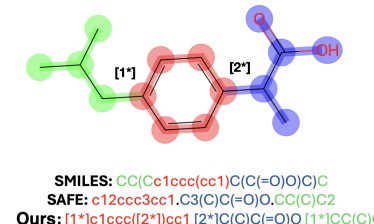

Figure 2: Comparison between SMILES, SAFE, and our notation for the same molecule. Our notation preserves fragment integrity while providing explicit attachment point numbering that facilitates bidirectional modeling of molecular structure.

molecule. The resulting fragmented SMILES strings are tokenized at the atomic level[4], yielding a vocabulary of 204 tokens.

## 2.3 FINETUNING

While *de novo* generation or fragment-constrained generation of small molecules is certainly an interesting and challenging topic, its practical utility in drug discovery is limited. A central contribution of our work is therefore the exploration of tasks with direct real world relevance, such as property optimization, docking score improvement, and lead refinement. These tasks require models not only

---

[4]Note that we do not tokenize e.g. 'Cl' as two tokens, but as a single one. The same holds for the attachment point tokens.

to generate chemically valid structures, but to navigate chemical space in a goal directed manner under realistic computational constraints. Because the finetuning strategies we employ are tightly coupled to the requirements of each experimental setting, we introduce them directly within the corresponding sections (Sec. 3.3 onwards).

## 3 EXPERIMENTS

We evaluate InVirtuoGen on four molecular design tasks: (i) *de novo* generation of diverse, synthesizable, drug-like molecules; (ii) fragment-constrained design with predefined scaffolds or pharmacophores; (iii) target property optimization on the PMO benchmark (Gao et al., 2022a), assessing sample efficiency and oracle performance; and (iv) lead optimization, optimizing docking scores under similarity, synthesizability and drug-likeness constraints.

For a fair comparison, we pretrain on the same datasets as GenMol (Lee et al., 2025) and SAFE-GPT (El Mesbahi and Noutahi, 2024): ZINC (Sterling and Irwin, 2015) and UniChem (Chambers et al., 2013), containing roughly one billion molecules.

The non-autoregressive formulation enables bidirectional attention, making the model well-suited for the inherently unordered representation. However, this also implies that we explicitly decouple sequence length from the generation process by factorizing the output distribution as

$$p_\theta(\mathbf{x}) = p(n)\, p_\theta(\mathbf{x} \mid n), \tag{5}$$

where $p(n)$ models the sequence length. To compare with GenMol, our base distribution is also chosen to be the empirical length distribution of ZINC250k, a curated subset of ZINC (Sterling and Irwin, 2015) containing synthesizable, drug-like compounds.

### 3.1 *De Novo* GENERATION

For drug discovery, generated molecules must be diverse, synthesizable, and drug-like. We evaluate these aspects with four metrics, following prior work on fragmented SMILES: *Validity* (fraction of valid SMILES), *Uniqueness* (fraction of unique valid molecules), *Diversity* (average Tanimoto distance of Morgan fingerprints (Polykovskiy et al., 2018; Rogers and Hahn, 2010)), and *Quality* (Lee et al., 2025) (fraction of valid, unique molecules with QED $\geq 0.6$ (Bickerton et al., 2012) and SA$\leq 4$ (Ertl and Schuffenhauer, 2009)). Quality and diversity are the main criteria, but they trade off against each other. As in GenMol, we tune this balance via the softmax temperature $T$ and a noise scale $r$ (modulating Gumbel noise[5]). During generation, $r$ is damped as $(1 - t)$ and $T$ is annealed, promoting early exploration and late refinement. Empirically, sampling directly from the predicted token-wise distribution

$$X_{t+h}^i \sim \hat{p}_t^i(X_t), \tag{6}$$

outperforms Eq. 2 significantly. Importantly, our update departs from masked discrete diffusion models: instead of replacing only masked tokens, all sequence positions possibly change simultaneously at every step, leading to a refinement process rather than arbitrary order completion. We provide more details and investigate the benefits of our sampling method in Appendix B.1.

**Results** In Figure 3, we present and compare our results to other state-of-the-art fragment-based generative models. Increasing the time-granularity (i.e., using smaller timestep sizes $h$) consistently improves both quality and diversity. InVirtuoGen consistently achieves a superior pareto frontier, with the largest gains at high time-granularity ($h = 0.001$), outperforming all baselines across the full quality-diversity spectrum. As described in Appendix B.1, sampling with $h = 0.01$ yields a comparable number of model calls, while still showing a modest performance increase over GenMol, particularly in the lower-diversity regime. We provide the ZINC250k sequence length distribution, non-curated samples, timing studies and additional results in Appendix B.1, including the pareto frontier obtained by sampling according to Eq. 2.

### 3.2 FRAGMENT-CONSTRAINED GENERATION

A core task in FBDD is to generate molecules given one or more fragments the molecule should contain. However, both next-token prediction models and masked discrete diffusion models such as GenMol can be prompted in a straightforward manner by using a prefix. In contrast, our model predicts a potential completely different sequence at every timestep. Thus strictly requiring the presence of the exact prompt significantly impacts the performance our model, since to ensure that the

---

[5]GenMol instead perturbs the order of the confidence-based token unmasking

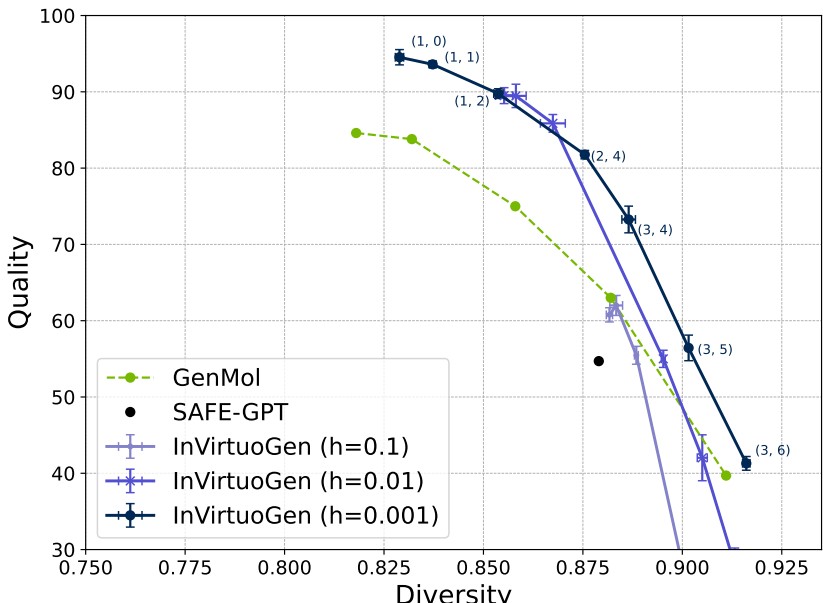

Figure 3: Quality-diversity trade-off for GenMol, SAFE-GPT (single point, as no quality-diversity scan data is available), and our model at different simulation time granularities ($h \in \{0.1, 0.01, 0.001\}$). Curves correspond to varying sampling noise $(T, r)$, where $T$ is the softmax temperature and $r$ is the Gumbel noise scale.

generated molecules remain consistent with given fragments, the corresponding positions are naively overwritten at every timestep of the simulation. We note that fragment-constrained generation is fundamentally at odds with our refinement philosophy. It requires fixing certain positions, preventing the holistic refinement that makes our approach powerful. Nevertheless, we include additional results in Appendix B.2. That said, prompting remains an important feature of our model. Unlike strict fragment enforcement, soft prompting enables controlled exploration in the neighborhood of an input molecule, making it a useful mechanism for guided search and local optimization, as we show in the following section.

## 3.3 TARGET PROPERTY OPTIMIZATION

The PMO benchmark evaluates molecular optimization under conditions that mirror practical drug discovery, with limited oracle calls and diverse pharmacologically relevant objectives. We introduce a hybrid approach that combines a genetic algorithm, which provides fast convergence through recombination of high-scoring molecules, with PPO-based reinforcement learning adapted to discrete flows, which enables gradient-guided refinement under shaped reward signals. Importantly, the combination of the two methodologies is simplified by our model's ability to accept full sequences as input. Note, that for this experiment we used the standard discrete flow sampling from Eq. 2, a short explanation for this choice is given in App. B.3.1.

**Genetic Algorithm** Because our model operates on full-length sequences, the initial state $x_{t=0}$ is constructed as a mixture of top-performing molecules. We maintain a vocabulary of high-scoring molecules with pairwise Tanimoto distance $\geq 0.7$ on Morgan fingerprints. To produce an offspring, two parents are sampled without replacement using rank-based probabilities $p(m) = 1/(\text{rank}(m) + \kappa M)$, where $M$ is the vocabulary size and $\kappa$ controls the contribution of lower-ranked entries. Parents are then decomposed by the fragmentation rule, and offspring are formed by replacing one fragment of the first parent with a fragment from the second, joined by a separator token in the fragmented SMILES space (i.e. simple string concatenation). GenMol joins fragments at fixed dummy attachment points, while our refinement method can adjust the local context and explore structural variants around the fragment. Naturally, most resulting offspring are not valid molecules, but they only provide the starting state $x_{t=0}$ for our model. While in traditional GA the offspring is mutated, we adopt the

mutation operators of Jensen (2019) and apply them to the best-performing molecules found so far to explore their neighborhood.

**Reinforcement Learning**    We adapt PPO (Schulman et al., 2017) to fine-tune the discrete flow policy. Unlike autoregressive models, where the policy log-probability is directly available as the sum of next-token log-likelihoods, discrete flows do not yield a tractable $\log p(x)$ over entire sequences. Instead, following the discrete flow matching framework of Gat et al. (2024), we approximate the log-probability via Monte Carlo estimation over perturbed states. For every sequence, we draw timesteps $t \sim U(0,1)$, apply the noise schedule to obtain a partially noised state $x_t$ and optimize the time-weighted loss

$$L = \frac{1}{1-t^2} \sum_{\text{noised positions}} \log \pi_\theta(x_1^i \mid x_t, t). \tag{7}$$

Our rewards are computed as $A = \frac{r-\bar{r}}{\sigma_r+\epsilon}$, where $r$ is the oracle scores, $\bar{r}$ denotes the batch mean, $\sigma_r$ the standard deviation and $\epsilon$ provides numerical stability. The clipped PPO surrogate $\rho(\theta) = \exp(\log \pi_\theta - \log \pi_{\text{old}})$ is optimized in the standard way.

**Adaptive Sequence Length Sampling**    As mentioned previously, our model requires a sequence length as input during generation. To accelerate convergence to well-performing molecule lengths, we employ an adaptive bandit that favors lengths with consistently high rewards while still retaining exploration through the prior distribution. We use a peak-seeking variant that blends best-so-far performance, reward quantiles, and an exploration bonus (see Appendix A.3.1).

**Combined Optimization Algorithm**    The overall procedure, combining GA exploration with PPO fine-tuning of the discrete flow, is summarized in Alg. 1, and additional implementation details are provided in Appendix A.3. Unlike GenMol or $f$-RAG, our method uses a single hyperparameter configuration across all tasks, highlighting that the performance gains arise from the algorithmic design rather than extensive hyperparameter tuning.

---

**Algorithm 1** Target-Property Optimization (GA + PPO)

---

**Require:** Model $\pi$, frozen prior $\pi_{\text{old}}$, oracle $O$, population $\mathcal{P}$, bandit $B$, maximum oracle calls $N_{\max}$, fragmentation rule $\mathcal{R}$, mutation op, PPO params $(\epsilon, c_{\text{neg}}, \beta)$
   **while** oracle calls $< N_{\max}$ **do**
      sample parent pairs from $\mathcal{P}$; draw lengths $\boldsymbol{\ell} \sim B$
      $\mathcal{P}_{\text{off}} \leftarrow \{X \sim \pi(\cdot \mid \text{crossover}(\mathcal{R}(p_1), \mathcal{R}(p_2)), \boldsymbol{\ell})\}$
      $\mathcal{P}_{\text{off}} \leftarrow \mathcal{P}_{\text{off}} \cup \{\text{mutate}(x) : x \in \text{top}_N(\mathcal{P})\}$
      $\mathbf{r} \leftarrow O(\mathcal{P}_{\text{off}})$
      **for** $k = 1$ to $K$ **do**
         $\theta \leftarrow \theta - \nabla_\theta L(\theta; \mathbf{r}, \pi_\theta, \pi_{\text{old}})$
      **end for**
      update $B$ with $(\boldsymbol{\ell}, \mathbf{r})$;   $\mathcal{P} \leftarrow \text{top}\{\mathcal{P} \cup \mathcal{P}_{\text{off}}\}$
      $\pi_{\text{old}} \leftarrow \pi$
   **end while**
   **return** top molecules

---

**Results**    The PMO benchmark comprises 23 single-objective molecular optimization tasks. Evaluation considers the achieved score and sample efficiency, summarized by the **top10 AUC** metric: the area under the curve of the mean score of the top ten molecules as a function of oracle calls. The scores are normalized to $[0, 1]$, and each run is limited to $10,000$ oracle evaluations. GenMol (Lee et al., 2025) and f-RAG (Lee et al., 2024) initialize their populations by screening the entire ZINC250k dataset, i.e. performing 250,000 additional oracle calls before optimization begins. Thus, while they nominally report results with 10k oracle calls, the effective budget is closer to 260,000. To ensure comparability, we evaluate our method under both setups: with prescreening on ZINC250k (Tab.1), directly comparable to GenMol and f-RAG, and without prescreening (Tab.2), directly comparable to baselines such as REINVENT (Olivecrona et al., 2017), MolGA (Tripp and Hernández-Lobato, 2023), and Genetic GFN (Kim et al., 2024), which do not use any prior oracle information. Due to space constraints, we only include up to three top-performing baselines, ranked by the sum of AUC-top10 scores across all benchmark tasks. In both setups, InVirtuoGen consistently achieves the

Table 1: Comparison of models on the PMO benchmark that screen ZINC250k before initialization. We report the AUC-top10 scores, averaged over three runs with standard deviations. Best results and those within one standard deviation of the best are indicated in bold. The scores for $f$-RAG (Lee et al., 2024) and GenMol (Lee et al., 2025) are taken from the respective publications.

| Oracle | InVirtuoGen | GenMol | f-RAG |
|---|---|---|---|
| albuterol similarity | **0.975** (± 0.016) | 0.937 (± 0.010) | **0.977** (± 0.002) |
| amlodipine mpo | **0.836** (± 0.031) | **0.810** (± 0.012) | 0.749 (± 0.019) |
| celecoxib rediscovery | **0.839** (± 0.013) | 0.826 (± 0.018) | 0.778 (± 0.007) |
| deco hop | **0.968** (± 0.012) | **0.960** (± 0.010) | 0.936 (± 0.011) |
| drd2 | **0.995** (± 0.000) | **0.995** (± 0.000) | 0.992 (± 0.000) |
| fexofenadine mpo | **0.904** (± 0.000) | 0.894 (± 0.028) | 0.856 (± 0.016) |
| gsk3b | **0.988** (± 0.001) | 0.986 (± 0.003) | 0.969 (± 0.003) |
| isomers c7h8n2o2 | **0.988** (± 0.002) | 0.942 (± 0.004) | 0.955 (± 0.008) |
| isomers c9h10n2o2pf2cl | **0.898** (± 0.018) | 0.833 (± 0.014) | 0.850 (± 0.005) |
| jnk3 | 0.898 (± 0.031) | **0.906** (± 0.023) | **0.904** (± 0.004) |
| median1 | 0.386 (± 0.003) | **0.398** (± 0.000) | 0.340 (± 0.007) |
| median2 | **0.377** (± 0.006) | 0.359 (± 0.004) | 0.323 (± 0.005) |
| mestranol similarity | **0.991** (± 0.002) | 0.982 (± 0.000) | 0.671 (± 0.021) |
| osimertinib mpo | **0.881** (± 0.012) | **0.876** (± 0.008) | 0.866 (± 0.009) |
| perindopril mpo | **0.753** (± 0.019) | 0.718 (± 0.012) | 0.681 (± 0.017) |
| qed | **0.943** (± 0.000) | 0.942 (± 0.000) | 0.939 (± 0.001) |
| ranolazine mpo | **0.854** (± 0.012) | 0.821 (± 0.011) | 0.820 (± 0.016) |
| scaffold hop | **0.711** (± 0.081) | 0.628 (± 0.008) | 0.576 (± 0.014) |
| sitagliptin mpo | **0.743** (± 0.022) | 0.584 (± 0.034) | 0.601 (± 0.011) |
| thiothixene rediscovery | 0.652 (± 0.024) | **0.692** (± 0.123) | **0.584** (± 0.009) |
| troglitazone rediscovery | 0.853 (± 0.003) | **0.867** (± 0.022) | 0.448 (± 0.017) |
| valsartan smarts | **0.935** (± 0.012) | 0.822 (± 0.042) | 0.627 (± 0.058) |
| zaleplon mpo | **0.624** (± 0.040) | **0.584** (± 0.011) | 0.486 (± 0.004) |
| **Sum** | **18.993** (± 0.219) | 18.362 | 16.928 |

best overall performance, considering the sum over all tasks. Ablations in Appendix B.3.1 show that all components of our optimization stack matter. In particular, PPO without prescreening and any genetic algorithm yields higher performance than REINFORCE, a common baseline used in industry.

## 3.4 LEAD OPTIMIZATION

Given an initial seed molecule, the goal in lead optimization is to generate leads that exhibit improved binding affinity to a target protein while satisfying constraints on the generated molecules. For the following experiments, the constraints are QED $\geq$ 0.6, SA $\leq$ 4, and Tanimoto similarity $\geq \delta$ to the seed molecule, where $\delta \in \{0.4, 0.6\}$. We follow the benchmark of Lee et al. (2025) and evaluate on five target proteins (parp1, fa7, 5ht1b, braf, jak2), each with three active ligands as molecule seeds. Performance is measured by the docking score (DS) of the most optimized lead (lower is better). We use the same optimization method as in the previous section, but we scale the docking score $DS$ by constraint satisfaction:

$$S(m) = \frac{DS}{15} \left(1 - \text{penalty}(\text{QED}(m), \text{SA}(m), \text{SIM}(m)\right). \tag{8}$$

where the penalty increases when QED < 0.6, SA > 4, or SIM < $\delta$.

**Results**  Table 3 compares our approach against GenMol, RetMol, and GraphGA. Our model consistently achieves competitive or superior docking scores across most proteins and similarity thresholds. Notably, our method remains effective even under the stricter $\delta = 0.6$ constraint, where baseline methods frequently fail to produce improved leads. For example, on parp1 and jak2, our model obtains substantially better docking scores than the baselines. The use of a soft-constraint oracle during training proves advantageous, allowing our model to explore chemical space more effectively while still converging to molecules that meet all constraints. While Tanimoto similarity is a standard proxy for structural similarity, it has known limitations as it reduces complex molecular relationships to a single fingerprint overlap score. To give a more complete view, we also report results without this constraint in Appendix B.4.

Table 2: The results of the best performing models on the PMO benchmark, where we quote the AUC-top10 averaged over 3 runs with standard deviations. The best results are highlighted in bold. Values within one standard deviation of the best are also marked in bold. The results for Genetic GFN (Kim et al., 2024) and Mol GA (Tripp and Hernández-Lobato, 2023) are taken from the respective papers. The other results are taken from the original PMO benchmark paper by (Gao et al., 2022a).

| Oracle | InVirtuoGen (no prescreen) | Gen. GFN | Mol GA | REINVENT |
|---|---|---|---|---|
| albuterol similarity | **0.950** $(\pm 0.017)$ | **0.949** $(\pm 0.010)$ | 0.896 $(\pm 0.035)$ | 0.882 $(\pm 0.006)$ |
| amlodipine mpo | 0.733 $(\pm 0.043)$ | **0.761** $(\pm 0.019)$ | 0.688 $(\pm 0.039)$ | 0.635 $(\pm 0.035)$ |
| celecoxib rediscovery | **0.798** $(\pm 0.028)$ | **0.802** $(\pm 0.029)$ | 0.567 $(\pm 0.083)$ | 0.713 $(\pm 0.067)$ |
| deco hop | **0.748** $(\pm 0.109)$ | **0.733** $(\pm 0.109)$ | **0.649** $(\pm 0.025)$ | **0.666** $(\pm 0.044)$ |
| drd2 | **0.985** $(\pm 0.002)$ | 0.974 $(\pm 0.006)$ | 0.936 $(\pm 0.016)$ | 0.945 $(\pm 0.007)$ |
| fexofenadine mpo | 0.845 $(\pm 0.016)$ | **0.856** $(\pm 0.039)$ | **0.825** $(\pm 0.019)$ | 0.784 $(\pm 0.006)$ |
| gsk3b | **0.952** $(\pm 0.016)$ | 0.881 $(\pm 0.042)$ | 0.843 $(\pm 0.039)$ | 0.865 $(\pm 0.043)$ |
| isomers c7h8n2o2 | **0.968** $(\pm 0.005)$ | **0.969** $(\pm 0.003)$ | 0.878 $(\pm 0.026)$ | 0.852 $(\pm 0.036)$ |
| isomers c9h10n2o2pf2cl | 0.874 $(\pm 0.013)$ | **0.897** $(\pm 0.007)$ | 0.865 $(\pm 0.012)$ | 0.642 $(\pm 0.054)$ |
| jnk3 | **0.825** $(\pm 0.016)$ | 0.764 $(\pm 0.069)$ | 0.702 $(\pm 0.123)$ | 0.783 $(\pm 0.023)$ |
| median1 | 0.342 $(\pm 0.008)$ | **0.379** $(\pm 0.010)$ | 0.257 $(\pm 0.009)$ | 0.356 $(\pm 0.009)$ |
| median2 | 0.288 $(\pm 0.008)$ | **0.294** $(\pm 0.007)$ | **0.301** $(\pm 0.021)$ | 0.276 $(\pm 0.008)$ |
| mestranol similarity | **0.797** $(\pm 0.033)$ | 0.708 $(\pm 0.057)$ | 0.591 $(\pm 0.053)$ | 0.618 $(\pm 0.048)$ |
| osimertinib mpo | **0.870** $(\pm 0.005)$ | 0.860 $(\pm 0.008)$ | 0.844 $(\pm 0.015)$ | 0.837 $(\pm 0.009)$ |
| perindopril mpo | **0.645** $(\pm 0.032)$ | 0.595 $(\pm 0.014)$ | 0.547 $(\pm 0.022)$ | 0.537 $(\pm 0.016)$ |
| qed | **0.942** $(\pm 0.000)$ | **0.942** $(\pm 0.000)$ | 0.941 $(\pm 0.001)$ | 0.941 $(\pm 0.000)$ |
| ranolazine mpo | **0.848** $(\pm 0.010)$ | 0.819 $(\pm 0.018)$ | 0.804 $(\pm 0.011)$ | 0.760 $(\pm 0.009)$ |
| scaffold hop | 0.589 $(\pm 0.021)$ | **0.615** $(\pm 0.100)$ | **0.527** $(\pm 0.025)$ | **0.560** $(\pm 0.019)$ |
| sitagliptin mpo | **0.709** $(\pm 0.029)$ | 0.634 $(\pm 0.039)$ | 0.582 $(\pm 0.040)$ | 0.021 $(\pm 0.003)$ |
| thiothixene rediscovery | **0.625** $(\pm 0.014)$ | 0.583 $(\pm 0.034)$ | 0.519 $(\pm 0.041)$ | 0.534 $(\pm 0.013)$ |
| troglitazone rediscovery | **0.595** $(\pm 0.053)$ | 0.511 $(\pm 0.054)$ | 0.427 $(\pm 0.031)$ | 0.441 $(\pm 0.032)$ |
| valsartan smarts | **0.210** $(\pm 0.297)$ | **0.135** $(\pm 0.271)$ | **0.000** $(\pm 0.000)$ | **0.178** $(\pm 0.358)$ |
| zaleplon mpo | 0.536 $(\pm 0.006)$ | **0.552** $(\pm 0.033)$ | **0.519** $(\pm 0.029)$ | 0.358 $(\pm 0.062)$ |
| **Sum** | **16.676** $(\pm 0.256)$ | 16.213 | 14.708 | 14.184 |

## 4 CONCLUSION

We have presented InVirtuoGen, a discrete flow-based generative model for fragmented SMILES. It is a versatile model employable in various stages of common practical drug discovery tasks allowing fragment-level control. By decoupling sequence length from token generation, we can show that a finer granularity during the simulation trajectory leads to more diverse and drug-like molecules in *de novo* generation. The uniform-source formulation also enables seamless integration with a genetic algorithm and PPO-motivated fine-tuning. Across *de novo* generation, fragment-constrained design, and target property optimization, our approach advances the state-of-the-art, achieving a new pareto frontier in quality-diversity trade-offs, with competitive quality and diversity in fragment-constrained tasks, a higher sum of top10 AUC in the PMO benchmark and similarly better results in lead optimization, where docking scores are optimized under constraints.

## 5 LIMITATIONS

Our fragmented SMILES representation discards stereochemistry, preventing modeling of stereospecific interactions. The rBRICS decomposition may miss chemically relevant fragmentation patterns, and the SA/QED metrics are coarse heuristics that poorly correlate with actual drug-likeness (Skoraczyński et al., 2023; Chen and Jung, 2024). Missing ADMET assessments limit our conclusions and all results remain proxy-based, requiring experimental validation. However, these issues are shared by all compared baselines.

Our sampling modification (Eq. 6) lacks theoretical justification despite strong empirical evidence. For fragment-constrained generation, we employ naive overwriting that disrupts the learned flow dynamics by forcing certain positions to remain fixed throughout the trajectory, contradicting the refinement paradigm central to our approach.

Table 3: Docking scores (lower is better) averaged over 3 random seeds. Bold indicates the best result per seed molecule. Values in parentheses indicate solutions with QED$> 0.6$ and SA$< 4$ that do not improve the docking score over the seed. For each seed molecule, its docking score, the quantitative estimate of drug-likeness and synthetic accessibility is given.

| Protein (DS/QED/SA) | $\delta = 0.4$ | | | | $\delta = 0.6$ | | | |
|---|---|---|---|---|---|---|---|---|
| | GenMol | RetMol | GraphGA | InVirtuoGen | GenMol | RetMol | GraphGA | InVirtuoGen |
| parp1 | | | | | | | | |
| -7.3/0.888/2.61 | -10.6 | -9.0 | -8.3 | **-14.1** $(\pm0.4)$ | -10.4 | - | -8.6 | **-12.3** $(\pm0.2)$ |
| -7.8/0.758/2.74 | -11.0 | -10.7 | -8.9 | **-13.4** $(\pm0.6)$ | -9.7 | - | -8.1 | **-11.7** $(\pm0.5)$ |
| -8.2/0.438/2.91 | **-11.3** | -10.9 | - | $-9.0$ $(\pm1.3)$ | -9.2 | - | - | **-10.7** $(\pm0.9)$ |
| fa7 | | | | | | | | |
| -6.4/0.284/2.29 | **-8.4** | -8.0 | -7.8 | **-8.4** $(\pm0.4)$ | **-7.3** | **-7.6** | **-7.6** | **-7.7** $(\pm0.4)$ |
| -6.7/0.186/3.39 | **-8.4** | - | -8.2 | **-8.9** $(\pm0.5)$ | **-7.6** | - | **-7.6** | $-7.5$ $(\pm0.3)$ |
| -8.5/0.156/2.66 | - | - | - | $(-8.0$ $(\pm0.3))$ | - | - | - | $(-7.4$ $(\pm0.4))$ |
| 5ht1b | | | | | | | | |
| -4.5/0.438/3.93 | -12.9 | -12.1 | -11.7 | **-13.3** $(\pm0.1)$ | **-12.1** | - | -11.3 | **-12.4** $(\pm0.5)$ |
| -7.6/0.767/3.29 | **-12.3** | -9.0 | -12.1 | $-12.0$ $(\pm0.7)$ | **-12.0** | -10.0 | **-12.0** | **-12.0** $(\pm0.4)$ |
| -9.8/0.716/4.69 | **-11.6** | - | - | $-10.9$ $(\pm0.2)$ | **-10.5** | - | - | **-10.6** $(\pm0.1)$ |
| braf | | | | | | | | |
| -9.3/0.235/2.69 | -10.8 | **-11.6** | -9.8 | $-10.1$ $(\pm0.0)$ | - | - | - | **-9.7** $(\pm0.1)$ |
| -9.4/0.346/2.49 | **-10.8** | - | - | **-10.8** $(\pm0.1)$ | -9.7 | - | - | **-10.4** $(\pm0.3)$ |
| -9.8/0.255/2.38 | -10.6 | - | **-11.6** | $-10.6$ $(\pm0.4)$ | **-10.5** | - | -10.4 | $-10.3$ $(\pm0.3)$ |
| jak2 | | | | | | | | |
| -7.7/0.725/2.89 | **-10.2** | -8.2 | -8.7 | **-10.2** $(\pm0.8)$ | -9.3 | -8.1 | - | **-9.7** $(\pm0.3)$ |
| -8.0/0.712/3.09 | -10.0 | -9.0 | -9.2 | **-10.5** $(\pm0.3)$ | -9.4 | - | -9.2 | **-10.4** $(\pm0.1)$ |
| -8.6/0.482/3.10 | -9.8 | - | - | **-10.2** $(\pm0.2)$ | - | - | - | **-10.3** $(\pm0.2)$ |
| Sum | -148.7 | -88.5 | -96.3 | **-152.4 (-160.4)** | -117.7 | -25.7 | -74.8 | **-145.7 (-153.1)** |

## 6 IMPACT, REPRODUCIBILITY & LLMS USAGE

**Impact** The method enables efficient fragment-based design with potential benefits for drug discovery but carries risks if misused for harmful molecule design.

**Reproducibility** We release model checkpoints, scripts, a Dockerfile, and instructions to replicate the experimental results.

**LLM Usage** We used language models for text polishing and LaTeX/coding assistance. However, all experiments, analyses, and results were designed and conducted by the authors.

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

# A    EXPERIMENTAL DETAILS

## A.1    IMPLEMENTATION DETAILS

The backbone model, adapted from Peebles and Xie (2022), has 36 layers[6], 12 heads, the hidden dimension is chosen as 768, and uses rotary positional embeddings  (Su et al., 2023). Training is performed for one epoch with batch size of 300. Sequences of similar length are bucketed together to reduce the padding per batch. The maximal number of tokens per batch is limited to $25,000$. The AdamW (Loshchilov and Hutter, 2019) optimizer, with $(\beta_1 = 0.99, \beta_2 = 0.999)$ is used with a learning rate of $10^{-4}$. Additionally, the learning rate is varied according to a linear warm-up cosine annealing scheduler.

## A.2    FRAGMENT-CONSTRAINED GENERATION PARAMETERS

The time-granularity of 0.01 is used together with our proposed sampling from Eq. 6. We see a slight performance increase when increasing the sampling time-granularity, as demonstrated in Appendix B.2.

## A.3    TARGET-PROPERTY OPTIMIZATION PARAMETERS

A smaller model is used in this setting, with 12 layers instead of 36. The population for the genetic algorithm, which provides the starting points of the trajectory $\{x_{t=0}\}$, is updated every 50 scored samples. Out of 80 generated samples, 20 are obtained by mutating the top 20 candidates generated so far, using the mutation operators from Jensen (2019). For rank-based sampling, we set $\kappa = 0.001$. For the calculation of the Tanimoto distance, we use Morgan fingerprints with 2048 bits and a radius of 2 nodes. The model is updated with PPO after every 100 scored SMILES. For each sequence, 50 timesteps $t \sim U(0, 1)$ are sampled to construct the training dataset $\{x_t\}$ by interpolating between a sequence of random tokens and the sampled values. During each epoch, 10 optimization steps are performed with a learning rate of $10^{-4}$. The clipping coefficient for PPO is set to $\eta = 0.2$.

When constructing the initial population by prescreening ZINC250K, no experience replay is used. Otherwise, an experience replay buffer of 300 samples is maintained.

### A.3.1    PEAK-SEEKER BANDIT

Our peak-seeking bandit, given in Alg. 2 is a heuristic that combines elements of UCB bandits (Auer et al., 2002) with quantile-based bandit updates (David and Shimkin, 2016).

---

**Algorithm 2** Peak-Seeker Bandit for Adaptive Length Sampling

---

**Require:** Candidate lengths $\{n_k\}$, prior $\pi^{(0)}$, quantile target $q$, learning rate $\eta_q$, weights $(w_{\text{best}}, w_{\text{quant}})$, bandwidth $\sigma$, exploration bonus $c$, temperature $\tau$, floor probability $\epsilon$

1: **while** generating molecules **do**
2:     For each arm $k$, track visit count $N_k$, best reward $b_k$, and running quantile estimate $\hat{q}_k$
3:     Compute score $s_k \leftarrow w_{\text{best}} b_k + w_{\text{quant}} \hat{q}_k + \text{UCB}(N_k) + \text{neighborhood}(L_{\text{best}}, n_k)$
4:     For unvisited arms: use prior $\pi^{(0)}$ as fallback
5:     Convert scores to probabilities $p_k \propto \exp(s_k/\tau)$, apply floor $\epsilon$, normalize
6:     Sample sequence length $L \sim \text{Categorical}(p)$
7:     Generate molecules of length $L$ and obtain reward $r$
8:     Update $N_k, b_k \leftarrow \max(b_k, r), \hat{q}_k$ by quantile SGD
9:     Update global best $(L_{\text{best}}, r_{\text{best}})$ if $r > r_{\text{best}}$
10: **end while**

---

---

[6]For the results in the fragment-constrained generation and target property/lead optimization a smaller model with 12 layers was used to improve speed and reduce memory use.

# B  ADDITIONAL EXPERIMENTAL RESULTS

## B.1  DE NOVO GENERATION

**Validity and Uniqueness versus Diversity.**    Since the *Quality* metric represents a combined summary of *Uniqueness* and *Validity*, here we report the individual relationships between Uniqueness, Validity, and Diversity, as shown in Figures 4.

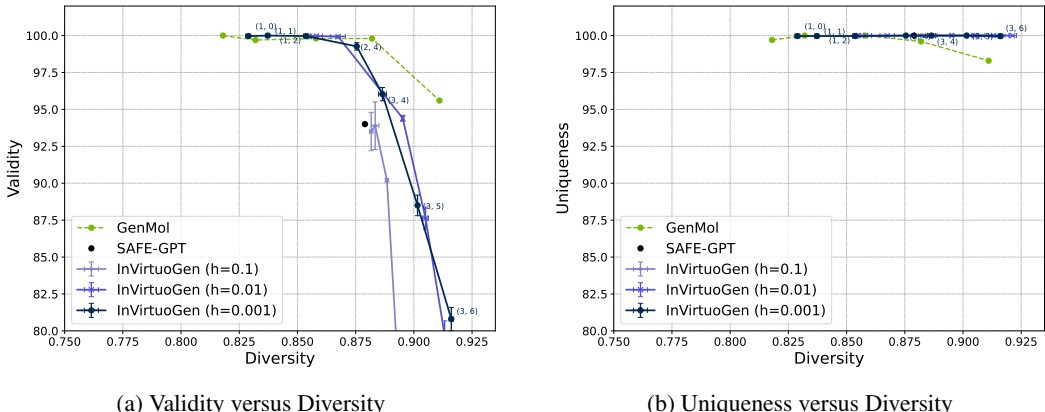

(a) Validity versus Diversity                     (b) Uniqueness versus Diversity

Figure 4: Trade-offs between Validity, Uniqueness, and Diversity in the generated molecules.

**Sequence Length Distribution**    In Fig. 5 we show the sequence length distribution of ZINC250K. The maximum observed length is 84, which implies that a masked model would require up to 84 steps to produce a sample. In contrast, our discrete flow model with a uniform source decouples the number of steps from the sequence length, allowing shorter sequences to be refined more within the same compute budget.

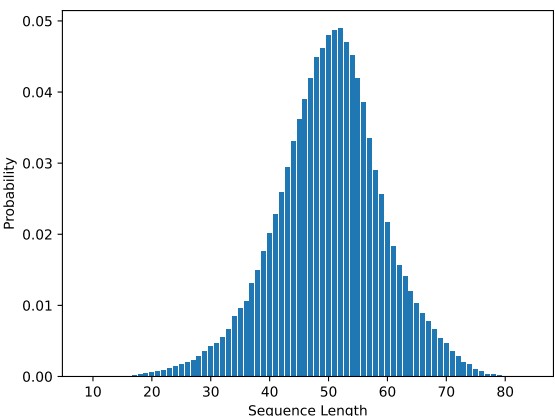

Figure 5: Sequence length distribution of ZINC250K. The maximum observed length is 84, which implies that a masked model requires at least 84 sampling steps, putting it close to our step size $h = 0.01$.

**Non-Curated Samples**    In Fig. 6 we provide non-curated samples from *de novo* generation with temperature $T = 1$ and randomness $r = 0$.

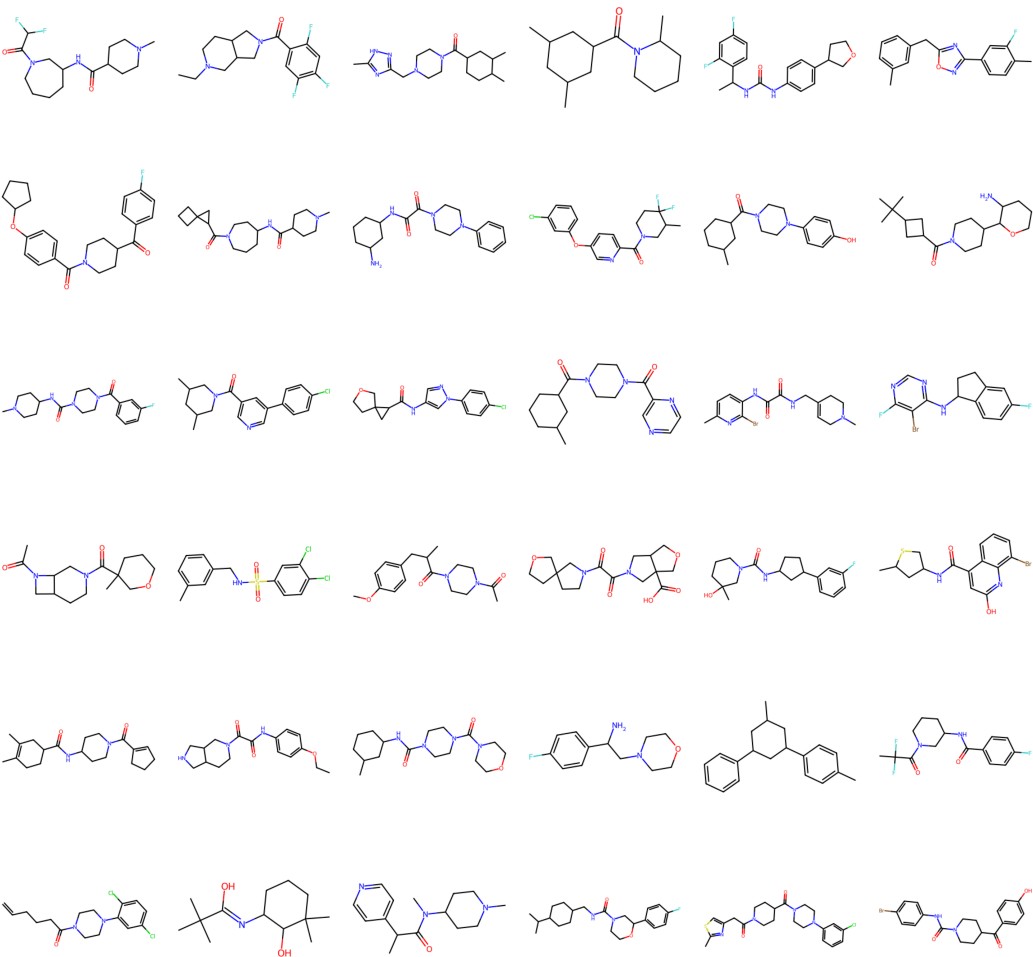

Figure 6: Non curated samples for *de novo* generation

**Comparison of Sampling Methods** In Fig. 7, we compare the quality versus diversity trade-off generated by sampling according to our proposed sampling method (Eq. 6) and for the one proposed by Gat et al. (2024) (Eq. 2). The benefit of increasing the sampling granularity disappears when standard sampling is used. In the following, we investigate possible causes for this.

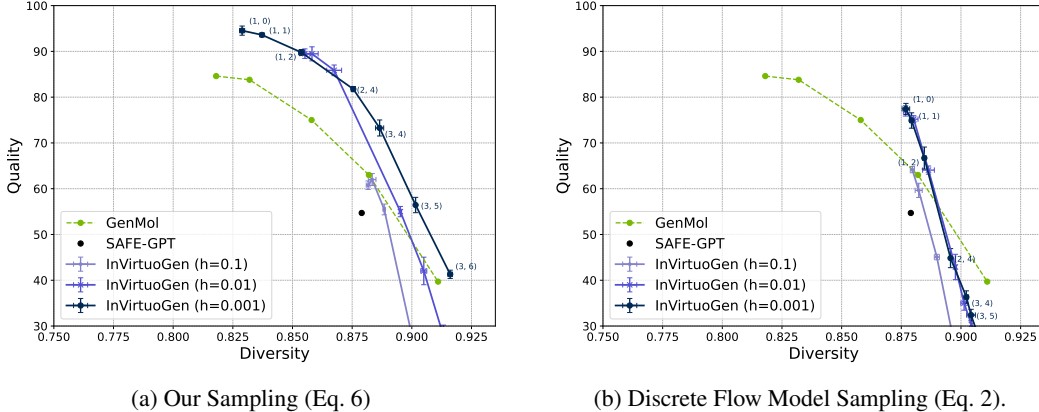

(a) Our Sampling (Eq. 6)  (b) Discrete Flow Model Sampling (Eq. 2).

Figure 7: Comparison of the different sampling methods in terms of quality vs diversity.

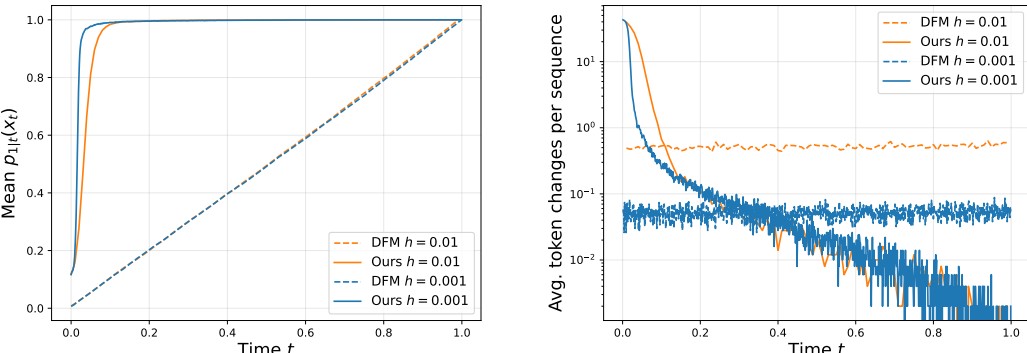

(a) Evolution of mean token probabilities $p_{1|T}(x_t)$.  (b) Average number of token changes per sequence.

Figure 8: Dynamic behavior during sampling. Our method rapidly concentrates token probabilities and progressively reduces the number of token changes, while standard sampling (DFM) increases confidence linearly and keeps changes nearly constant.

We provide statistics aggregated from the sampling trajectories. As shown in Fig. 8a, for our sampling method the probability density of the current tokens $p_{1|T}(x_t)$ quickly saturates at 1, while for standard sampling (Eq.2) it increases almost linearly across steps. This difference is mirrored in the dynamics of token changes (Fig. 8b): our approach begins with many parallel updates that gradually decay into a refinement phase, whereas standard sampling maintains a nearly constant but low rate of changes throughout. Our decay pattern is closer to the intuitive notion of refinement, where the number of modifications decreases as the sequence converges towards a high-probability solution. Finally, the cumulative number of token changes (Fig. 9) highlights a fundamental distinction: DFM sampling yields a step-size-invariant total number of modifications, effectively fixing the update budget, while our method scales with the granularity $h$, allowing more structured refinements when smaller steps are used. The observation that the number of token changes is independent of time granularity for standard DFM sampling provides a plausible explanation for the lack of improvement in standard sampling at higher resolutions.

Additionally, Figs. 10a and 11b show QED and SA distributions across time resolutions $h$. Under our sampling (Eq. 6), the QED distribution shifts markedly upward to higher values, while SA also seems to move to smaller values (Figs. 10a, 10b). In contrast, the standard discrete flow update (Eq. 2) more closely follows ZINC250k for both metrics (Figs. 11a, 11b). This upward shift in QED and downward shift in SA under our sampling method is consistent with the stronger quality–diversity

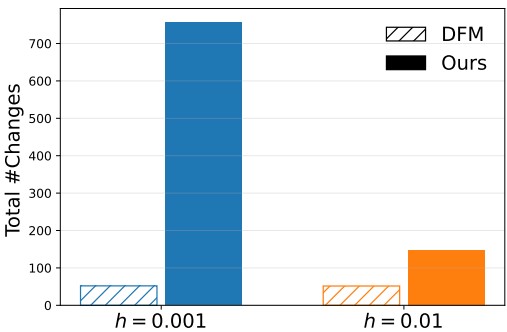

Figure 9: Cumulative number of token changes per sequence for different time resolutions $h$. DFM sampling shows step-size invariance, while our method scales the number of refinements with stepsize $h$.

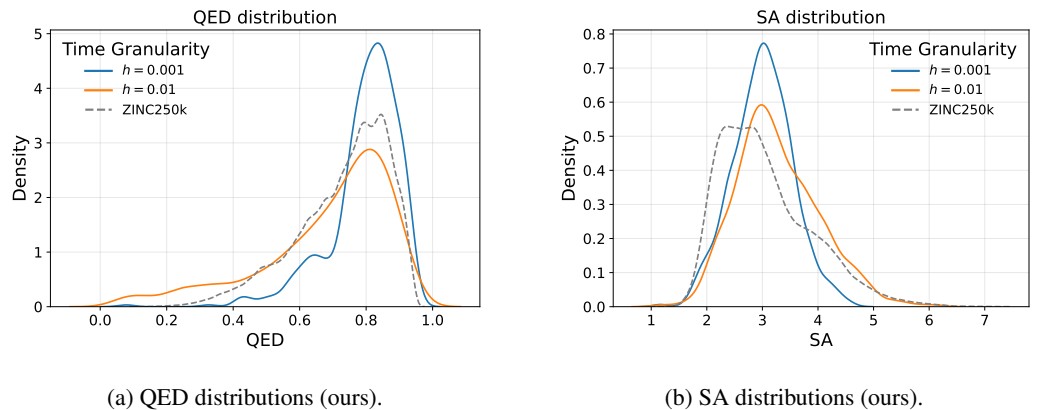

(a) QED distributions (ours).

(b) SA distributions (ours).

Figure 10: Distributions of QED and SA for molecules generated with our sampling rule (Eq. 6) at different time resolutions $h$. ZINC250k is shown as a dashed baseline.

frontier reported earlier. Although one might expect the generated distributions to overlap with ZINC250k, it is important to emphasize that ZINC250k contributed only a small fraction of the training data ($2.5 \times 10^5$ molecules vs. $10^9$ in total). The training dataset contains biological compoinds, which tend to be longer and complex and therfore exhibit higher synthetic accessibility. Moreover, the objective of our framework is not to reproduce the underlying training distribution but to generate drug-like molecules with optimized properties. Figure 12 compares standard sampling (Eq. 2) and

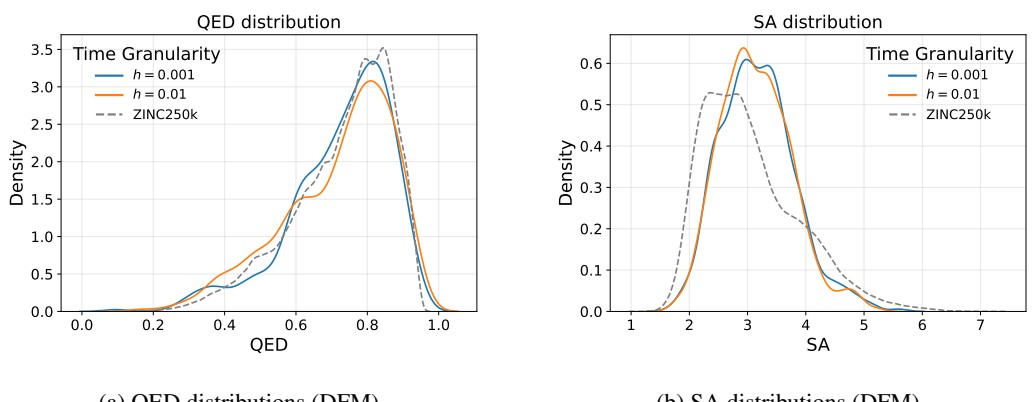

(a) QED distributions (DFM).

(b) SA distributions (DFM).

Figure 11: Distributions of QED and SA for molecules generated with the standard discrete flow update (Eq. 2) across different time resolutions $h$. ZINC250k is shown as a dashed baseline.

our sampling (Eq. 6) at ($T = 1$, $r = 0$). The left panel shows the average number of carbon atoms per sequence. Our sampling results in a slightly higher carbon frequency, though the difference is modest.

The right panel reports the distribution of the number of fragments per sequence, estimated from the maximal attachment index observed. Standard sampling produces a higher fraction of sequences with many fragments, whereas our sampling shifts weight toward intermediate fragment counts. We hypothesize that this structural difference translates into improved drug-likeness, but further investigation is necessary.

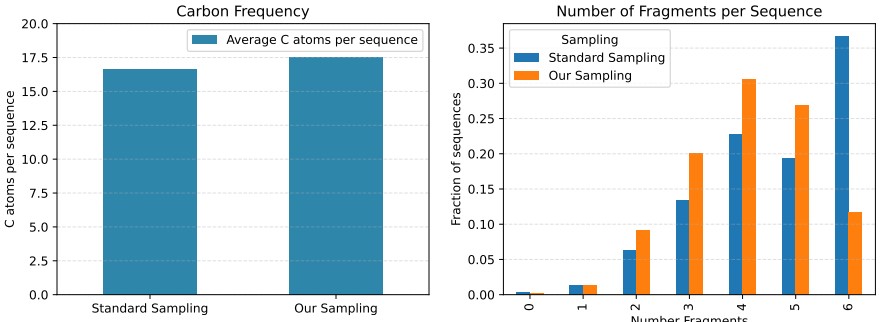

Figure 12: Sampling comparison at temperature $T = 1$ and noise $r = 0$. Left: average number of carbon atoms per sequence. Right: distribution of the number of fragments per sequence.

**Impact of Time-Weighting Loss**   In Fig. 13, we show results obtained by training the same model as for the other *de novo* generation experiments, but without time-weighting the loss terms. While the performance under sampling with Eq. 2 is nearly unchanged, the results with Eq. 6 are significantly worse without time-weighting, showing that our modification has a significant impact on the result.

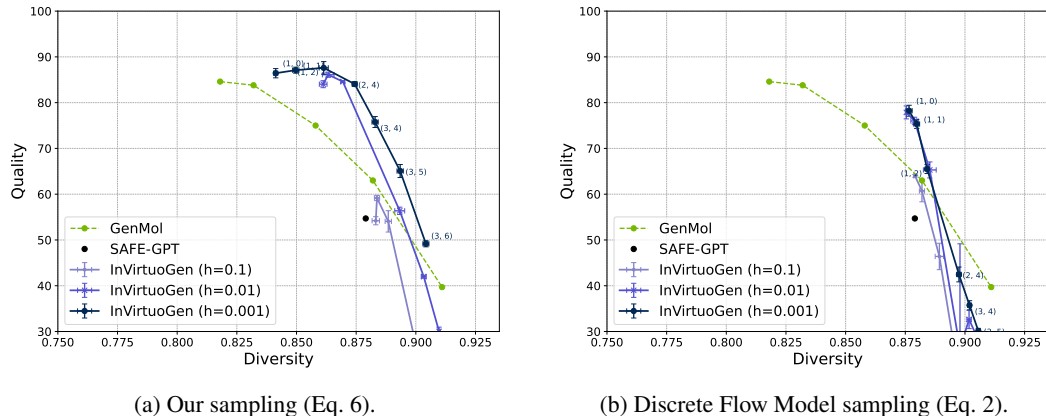

(a) Our sampling (Eq. 6).          (b) Discrete Flow Model sampling (Eq. 2).

Figure 13: Quality-diversity trade-off without using the time-weighting $\frac{1}{1-t^2}$ of the loss, given in Eq. 4. Left: results from sampling with Eq. 6. Right: results from sampling with Eq. 2.

**Timing Studies**   To compare with GenMol, we also report results for the smaller 12-layer model used in the target-property optimization section. Its quality-diversity frontier is shown in Fig. 14. Sampling with time granularity $h = 0.01$ on an RTX 4090 yields $20.2 \pm 0.3\,\text{s}$ for 1000 samples. Using the same setup with GenMol (instantiated from the provided configuration rather than a released checkpoint) we obtain $33.2 \pm 0.3\,\text{s}$. We emphasize, however, that speed is a minor factor in drug discovery, since downstream steps such as docking or wet-lab experiments typically dominate runtime.

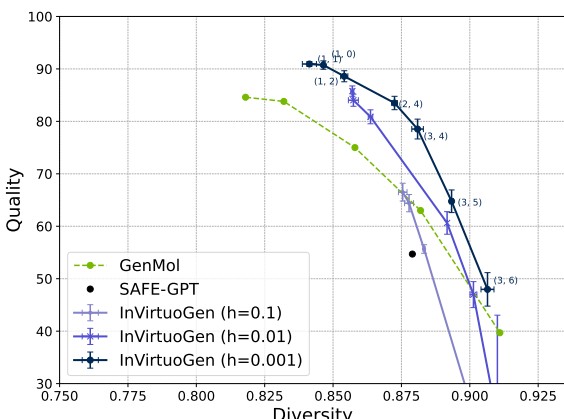

Figure 14: Quality-diversity frontier for the 12-layer model. At $h = 0.01$, InVirtuoGen achieves higher maximum quality to GenMol while attaining substantially higher diversity in the high-quality regime.

## B.2 FRAGMENT-CONSTRAINED GENERATION

We follow the benchmark of Noutahi et al. (2023), which uses fragments from ten known drugs, and evaluate five subtasks: linker design (connecting two or more terminal fragments with a feasible linker), scaffold morphing (modifying the core scaffold while preserving pharmacophoric features), motif extension (growing a fixed motif with new substituents), scaffold decoration (attaching functional groups at predefined positions), and superstructure generation (assembling multiple fragments into a coherent larger molecule). In our notation, as for GenMol, linker design and scaffold morphing yield identical prompts and therefore identical results.

**Updated comparison to prior work.** In contrast to the results reported in the original GenMol paper, our analysis of the public implementation shows that the official fragment-constrained results for linker design and scaffold morphing are produced through a multi-step rejection-filtering procedure. This procedure discards invalid, unparsable, or non-matching generations and therefore inflates the observed quality and validity. To ensure a fair comparison, we instead report the more relevant *single-step* linker-design results released in the GenMol repository update, which do not rely on this filtering. We include these values for completeness and comparability with prior work only.

In Table 4, we present the fragment-constrained generation results for all methods. Averaging over all five tasks and three random seeds, our model achieves competitive performance in terms of both quality and diversity. While the validity is lower across all tasks, we emphasize that validity in isolation is not the most informative metric in this benchmark. Validity and uniqueness contribute symmetrically to the quality metric, which more effectively captures the tradeoff between producing valid molecules and producing diverse ones. This avoids degenerate cases, such as models that repeatedly generate the same valid molecule. Moreover, lower validity can be compensated by generating additional samples, whereas low diversity or low quality cannot. For these reasons, we regard quality as the more practically meaningful metric. Finally, we note that GenMol tunes its generation parameters separately for each task, whereas our method uses a single unified parameter setting across all tasks.

Table 4: Performance across five fragment-constrained generation tasks, averaged over three random seeds: motif extension, linker design, superstructure generation, scaffold morphing, and scaffold decoration. In our setup, similar as for GenMol, scaffold decoration is identical to linker design, so results are shared.

| Task | Method | Diversity | Quality | Uniqueness | Validity |
|---|---|---|---|---|---|
| Motif Extension | SAFE-GPT | $0.56 \pm 0.003$ | $18.60 \pm 2.100$ | $66.80 \pm 1.200$ | $\mathbf{96.10 \pm 1.900}$ |
| | GenMol | $\mathbf{0.62 \pm 0.002}$ | $30.10 \pm 0.400$ | $77.50 \pm 0.100$ | $82.90 \pm 0.100$ |
| | InVirtuoGen | $\mathbf{0.62 \pm 0.005}$ | $\mathbf{39.27 \pm 1.078}$ | $\mathbf{96.83 \pm 0.290}$ | $68.97 \pm 0.759$ |
| Linker Design | SAFE-GPT | $\mathbf{0.55 \pm 0.007}$ | $21.70 \pm 1.100$ | $82.50 \pm 1.900$ | $\mathbf{76.60 \pm 5.100}$ |
| | GenMol | $0.53 \pm 0.002$ | $4.30 \pm 0.400$ | $\mathbf{97.80 \pm 0.500}$ | $16.70 \pm 0.2$ |
| | InVirtuoGen | $0.52 \pm 0.004$ | $\mathbf{22.33 \pm 1.250}$ | $84.76 \pm 1.620$ | $60.37 \pm 0.573$ |
| Scaffold Morphing | SAFE-GPT | $0.51 \pm 0.011$ | $16.70 \pm 2.300$ | $70.40 \pm 5.700$ | $58.90 \pm 6.800$ |
| | GenMol | $\mathbf{0.53 \pm 0.002}$ | $4.30 \pm 0.400$ | $\mathbf{97.80 \pm 0.500}$ | $16.70 \pm 0.2$ |
| | InVirtuoGen | $0.52 \pm 0.004$ | $\mathbf{22.33 \pm 1.250}$ | $84.76 \pm 1.620$ | $\mathbf{60.37 \pm 0.573}$ |
| Superstructure Design | SAFE-GPT | $0.57 \pm 0.028$ | $14.30 \pm 3.700$ | $83.00 \pm 5.900$ | $95.70 \pm 2.000$ |
| | GenMol | $0.60 \pm 0.009$ | $\mathbf{34.80 \pm 1.000}$ | $83.60 \pm 1.000$ | $\mathbf{97.50 \pm 0.900}$ |
| | InVirtuoGen | $\mathbf{0.73 \pm 0.001}$ | $27.43 \pm 0.953$ | $\mathbf{99.41 \pm 0.157}$ | $75.70 \pm 0.898$ |
| Scaffold Decoration | SAFE-GPT | $0.57 \pm 0.008$ | $10.00 \pm 1.400$ | $74.70 \pm 2.500$ | $\mathbf{97.70 \pm 0.300}$ |
| | GenMol | $\mathbf{0.59 \pm 0.001}$ | $31.80 \pm 0.500$ | $82.70 \pm 1.800$ | $96.60 \pm 0.800$ |
| | InVirtuoGen | $0.56 \pm 0.003$ | $\mathbf{36.37 \pm 1.096}$ | $\mathbf{88.58 \pm 1.130}$ | $90.70 \pm 0.616$ |
| **Average** | SAFE-GPT | $0.55 \pm 0.006$ | $16.26 \pm 1.031$ | $75.48 \pm 1.773$ | $\mathbf{85.00 \pm 1.788}$ |
| | GenMol | $0.57 \pm 0.002$ | $21.06 \pm 0.263$ | $87.88 \pm 0.436$ | $62.08 \pm 0.242$ |
| | InVirtuoGen | $\mathbf{0.59 \pm 0.001}$ | $\mathbf{29.55 \pm 0.813}$ | $\mathbf{90.87 \pm 0.445}$ | $71.22 \pm 0.399$ |

In Fig. 15 we provide non-cherry picked samples for the fragment-constrained generation task. The left-most figure in every row depicts the starting fragment(s).

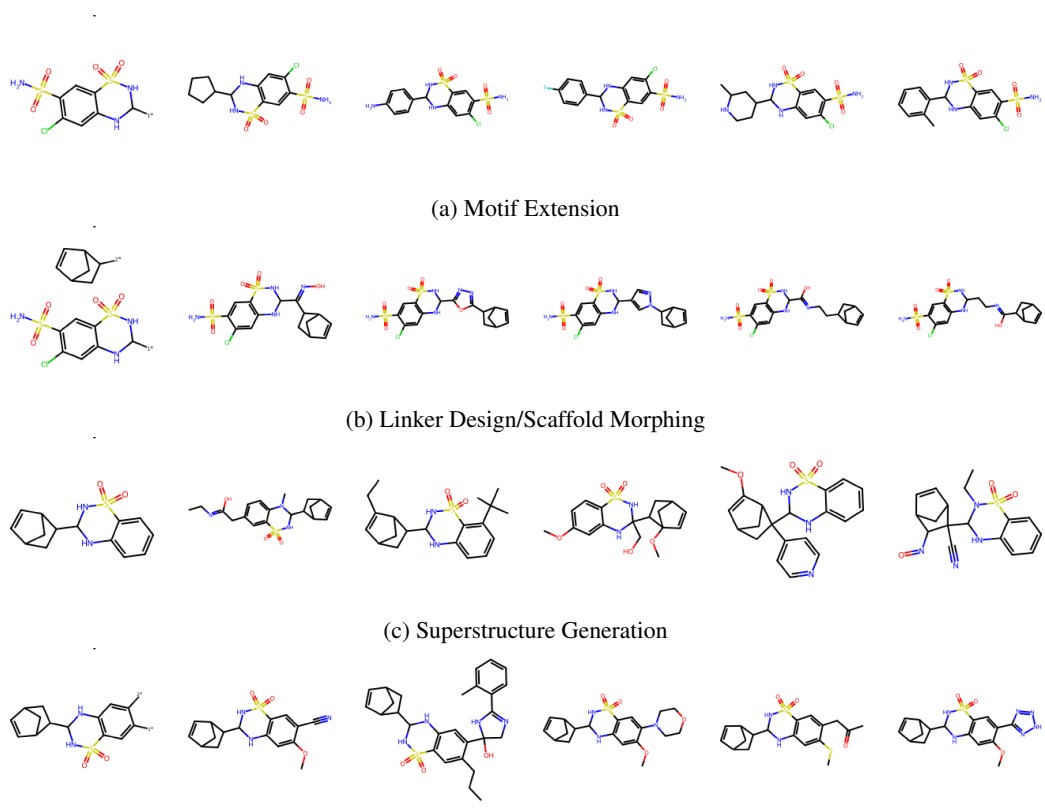

(a) Motif Extension

(b) Linker Design/Scaffold Morphing

(c) Superstructure Generation

(d) Scaffold Decoration

Figure 15: Non cherry-picked samples generated by InVirtuoGen on fragment-constrained design tasks. The left-most figure in every row depicts the starting fragment(s).

### B.3 TARGET-PROPERTY OPTIMIZATION

### B.3.1 ABLATION STUDIES

In Table 5 we show the results of ablation studies of the core components of our optimization framework. We include results for the following ablations:

- Including the experience replay in the prescreened setting.
- Sampling according to eq. 6 but with start time $t_{start} = 0.2$.
- Sampling according to eq. 6.
- Sampling sequence lengths from the Zinc250K distribution instead of using our Peak-Finder Bandit.
- No mutation applied to the best performing molecules.
- No PPO, relying only on the genetic algorithm.
- No prescreening, but leaving out the experience replay, yielding a significantly lower performance.
- A baseline without GA, mutation, or prescreening, which allows a fair comparison to REINVENT (Olivecrona et al., 2017) and slightly better performance.

As the table shows, the results obtained with our sampling method depend strongly on the start time of the trajectory simulation. We attribute this sensitivity to the dynamics in Fig. 8b: at early times many positions are simultaneously updated, which can diminish the performance gains provided by the genetic algorithm component. And because we did not want to tune our parameters, we stuck to sampling according to Eq 2.

Table 5: Ablation study results on the PMO benchmark. We report the AUC-top10 scores from single runs. Best results are highlighted in bold.

| Oracle | With Experience Replay | Sampling with eq. 6, $t_{start} = 0.2$ | Sampling with eq. 6 | No Bandit | No Mutation | No PPO | No Prescreen, No Experience Replay | No Prompter, No Mutation, No Prescreen |
|---|---|---|---|---|---|---|---|---|
| albuterol similarity | **0.993** | 0.975 | 0.972 | 0.991 | 0.928 | 0.881 | 0.969 | 0.851 |
| amlodipine mpo | **0.813** | 0.795 | 0.801 | 0.764 | 0.777 | 0.755 | 0.679 | 0.547 |
| celecoxib rediscovery | **0.872** | 0.866 | 0.795 | 0.785 | 0.758 | 0.815 | 0.834 | 0.768 |
| deco hop | 0.960 | 0.965 | 0.973 | **0.988** | 0.970 | 0.943 | 0.651 | 0.659 |
| drd2 | 0.995 | 0.995 | 0.995 | **0.995** | 0.986 | 0.986 | 0.984 | 0.950 |
| fexofenadine mpo | 0.908 | 0.870 | 0.879 | **0.913** | 0.834 | 0.853 | 0.852 | 0.735 |
| gsk3b | 0.990 | 0.989 | 0.987 | **0.992** | 0.976 | 0.972 | 0.949 | 0.867 |
| isomers c7h8n2o2 | **0.989** | 0.986 | 0.986 | 0.978 | 0.967 | 0.974 | 0.971 | 0.857 |
| isomers c9h10n2o2pf2cl | 0.876 | 0.882 | 0.871 | **0.919** | 0.851 | 0.834 | 0.888 | 0.798 |
| jnk3 | 0.892 | 0.917 | 0.881 | **0.917** | 0.807 | 0.889 | 0.825 | 0.701 |
| median1 | 0.382 | 0.379 | 0.370 | **0.385** | 0.370 | 0.363 | 0.342 | 0.308 |
| median2 | 0.333 | **0.386** | 0.367 | 0.377 | 0.331 | 0.363 | 0.304 | 0.241 |
| mestranol similarity | **0.992** | 0.986 | 0.986 | 0.983 | 0.963 | 0.962 | 0.723 | 0.764 |
| osimertinib mpo | **0.878** | 0.868 | 0.870 | 0.868 | 0.857 | 0.858 | 0.864 | 0.802 |
| perindopril mpo | 0.733 | **0.753** | 0.696 | 0.710 | 0.695 | 0.682 | 0.627 | 0.463 |
| qed | 0.943 | 0.943 | 0.943 | **0.944** | 0.943 | 0.943 | 0.943 | 0.941 |
| ranolazine mpo | **0.878** | 0.840 | 0.843 | 0.866 | 0.807 | 0.816 | 0.840 | 0.762 |
| scaffold hop | 0.654 | 0.632 | 0.648 | 0.657 | **0.790** | 0.595 | 0.619 | 0.522 |
| sitagliptin mpo | **0.772** | 0.766 | 0.613 | 0.738 | 0.457 | 0.546 | 0.693 | 0.324 |
| thiothixene rediscovery | **0.685** | 0.625 | 0.542 | 0.625 | 0.588 | 0.526 | 0.620 | 0.465 |
| troglitazone rediscovery | **0.870** | 0.859 | 0.812 | 0.855 | 0.833 | 0.836 | 0.434 | 0.418 |
| valsartan smarts | 0.870 | 0.906 | 0.891 | **0.927** | 0.766 | 0.715 | 0.000 | 0.150 |
| zaleplon mpo | 0.617 | **0.654** | 0.643 | 0.605 | 0.576 | 0.653 | 0.520 | 0.458 |
| **Sum** | **18.893** | 18.836 | 18.364 | 18.782 | 17.831 | 17.758 | 16.131 | 14.349 |

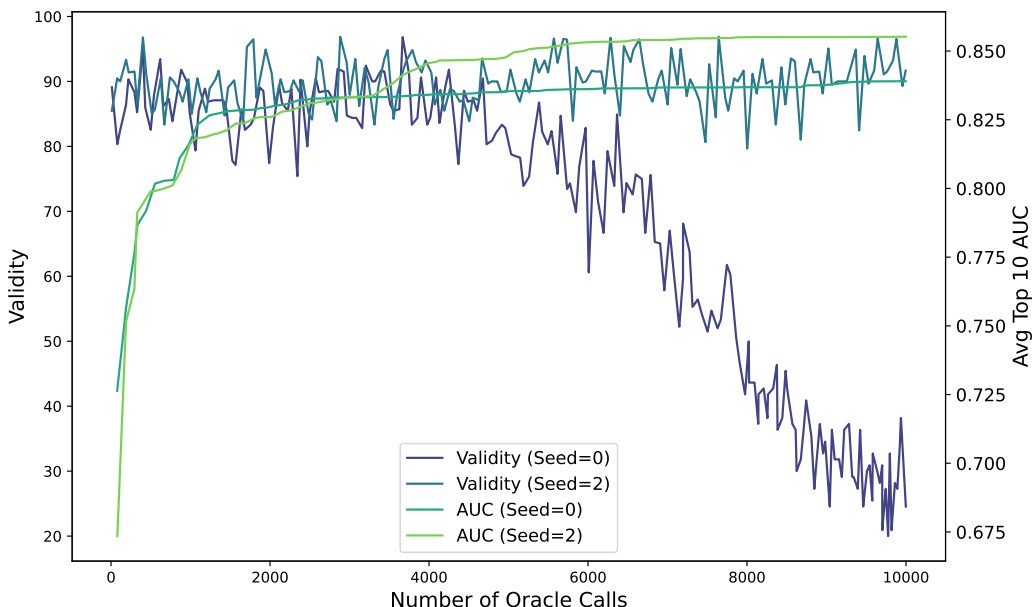

Figure 16: Validity versus Number Oracle Calls

### B.3.2 VALIDITY VS ORACLE CALLS

In Fig. 16, we illustrate the potential instability of our optimization routine. The plot shows both the fraction of valid samples and the top-10 AUC as functions of the number of oracle calls. We report two runs with identical configurations that differ only in their random seed. While one run remains stable throughout optimization, the other exhibits signs of policy collapse. Interestingly, the collapsing run achieves higher top-10 AUC, a pattern we repeatedly observed. We attribute this behavior to an overly exploitative optimization regime, and addressing this—potentially through stabilizing techniques such as proposed by Xi et al. (2025), is an important direction for future work.

### B.4 LEAD OPTIMIZATION

We provide the results obtained when not requiring similarity to the seed molecule in Tab. 6, showing that performance improves significantly when this restriction is removed. Constraints on drug-likeness and synthesizability are still enforced.

Table 6: Docking scores (lower is better) averaged over 3 seeds. Bold indicates the best result per seed. For each seed molecule, its docking score, the quantitative estimate of drug-likeness and synthetic accessibility is given.

| Protein (DS/QED/SA) | No sim InVirtuoGen |
|---|---|
| 5ht1b | |
| -4.5/0.438/3.93 | **-12.5** (±1.0) |
| -7.6/0.767/3.29 | **-13.4** (±0.5) |
| -9.8/0.716/4.69 | **-13.0** (±0.3) |
| braf | |
| -9.3/0.235/2.69 | **-12.5** (±0.2) |
| -9.4/0.346/2.49 | **-12.3** (±0.8) |
| -9.8/0.255/2.38 | **-12.2** (±0.4) |
| fa7 | |
| -6.4/0.284/2.29 | **-9.9** (±0.4) |
| -6.7/0.186/3.39 | **-10.2** (±0.5) |
| -8.5/0.156/2.66 | **-9.4** (±0.2) |
| jak2 | |
| -7.7/0.725/2.89 | **-12.0** (±0.3) |
| -8.0/0.712/3.09 | **-12.1** (±0.2) |
| -8.6/0.482/3.10 | **-11.6** (±0.7) |
| parp1 | |
| -7.3/0.888/2.61 | **-13.5** (±0.2) |
| -7.8/0.758/2.74 | **-13.3** (±0.7) |
| -8.2/0.438/2.91 | **-13.5** (±0.5) |
| Sum | **-181.4** |

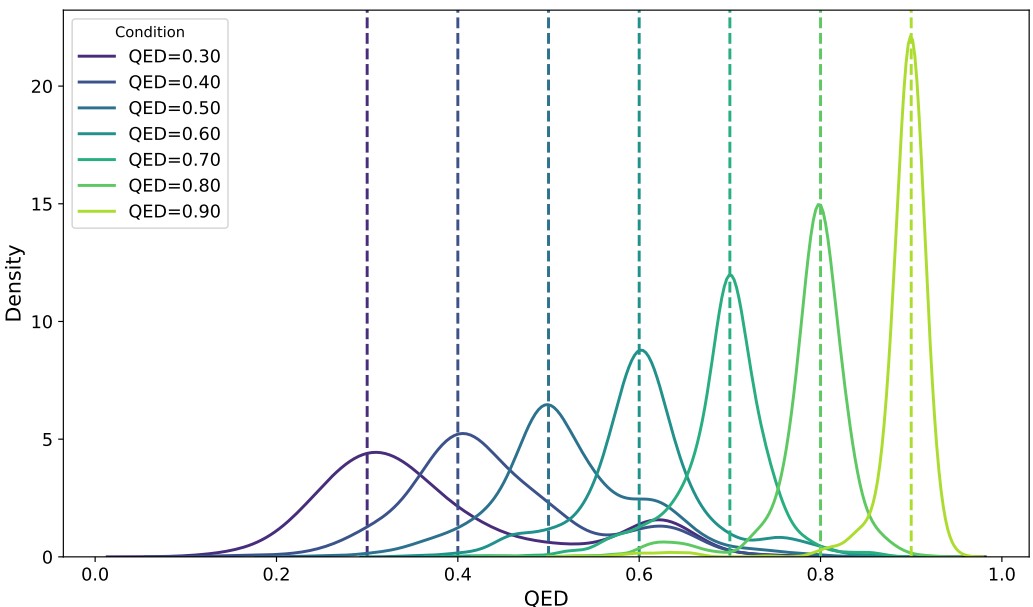

Figure 17: QED Distribution for QED-Conditioned Samples

### B.5 CONDITIONAL GENERATION VERSUS TARGET PROPERTY OPTIMIZATION

Here we briefly clarify the conceptual and practical differences between conditional generation and our target-property optimization framework. Conditional generation provides a model with an explicit property value (for example, a desired LogP or QED level) as an additional input during training as prior work explored (Lim et al., 2020; Seo et al., 2023). In our implementation, the condition is passed through a small neural module and added to the timestep embedding. Recent advances further introduce explicit guidance mechanisms for discrete flow models (Nisonoff et al., 2025), which could potentially improve the results adherance further. Figure 17 depicts the results obtained from fine-tuning our model with conditional inputs. The model produces QED distributions that shift consistently with the provided conditioning input, confirming that the model can be guided toward desired property regimes.

