# OpenReview forum: "Refine Drugs, Don’t Complete Them: Uniform-Source Discrete Flows for Fragment-Based Drug Discovery"
_ICLR.cc/2026/Conference — ICLR 2026 Poster_

### Official Review · Reviewer_eGk6 · 2025-10-28

**Soundness:** 1
**Presentation:** 1
**Contribution:** 1
**Rating:** 2
**Confidence:** 4

**Summary:**

Through this paper, the authors propose InVirtuoGen, a discrete flow generative model for fragmented SMILES for de novo and fragment-constrained generation, and target-property/lead optimization of small molecules. Specifically, InVirtuoGen learns to transform a uniform source over all possible tokens into the data distribution by refining the sequence.

**Strengths:**

- The authors provided the codebase.

**Weaknesses:**

I will combine the *Weaknesses* section and the *Questions* section. My concerns are as follows:
- The main weakness of this paper is its weak novelty. The method section is completely missing. The proposed InVirtuoGen framework is a combination of a discrete flow model (Section 2.1) and the fragmented SMILES notation (Section 2.2). Only the latter corresponds to the invention of this work, but its contribution appears to be very minor.  The new notation is very similar to the SAFE notation [1], and its advantages over the SAFE notation is not described. The sentence “Our notation preserves fragment integrity while providing explicit attachment point numbering” is included in the caption of Figure 2, but this also applies for the SAFE notation. Overall, I am not convinced that this work provides a significant contribution from an ML perspective compared to previous methods in the domain.
- The proposed InVirtuoGen framework shows very low molecular validity in the fragment-constrained generation tasks (Table 1).

---

**References:**

[1] Noutahi et al., Gotta be safe: A new framework for molecular design, ArXiv, 2023.

**Questions:**

Please see the *Weaknesses* section for my main concerns.

---

> ### Author Response · Authors · 2025-11-14
> **Addressing Weaknesses Pt.1**
>
> We thank you for acknowledging our efforts to provide a complete and reproducible code base! Although this was the only strength explicitly noted in your review, we hope the clarifications below help convey the broader set of contributions that our work brings.
> > **W1: The main weakness of this paper is its weak novelty.  The method section is completely missing.**
>
> While we agree that the original submission did not follow a strict method, experiment, and results separation, this was an intentional choice driven by the fact that many of our methodological contributions are tightly coupled to the practical tasks. In the revised version, we reorganized our contribution to explicitly denote this.
> We thank the reviewer for this comment and would like to take this opportunity to clarify our contributions more explicitly
>
> **1. How Our Refinement-Based Discrete Flow Model Differs Fundamentally from Prior Approaches**
> Our work is, to the best of our knowledge, the first to adapt discrete flow matching with a uniform source for molecular generation, where the entire token sequence is updated jointly at every step (We discuss the differences of standard discrete flow sampling and our approach in App.B1. This is fundamentally different from both autoregressive models (SAFE GPT) and masked discrete diffusion (GenMol). The use of a uniform source distribution is novel too and plays a key role in the optimization routine. Put shortly, it leads to major differences in training and sampling dynamics: while in masked discrete diffusion models, loss is only calculated on masked tokens and unmasked tokens remain unchanged during sampling, our approach is significantly different:
> the loss is calculated on all tokens, forcing the model to account for correctness over every token.
> For masked models, the number of sampling steps is directly related to the sequence length, as a token can only be unmasked once - i.e. the number of transitions between tokens is bounded by the sequence length. We address this in App. B.1
>
> **2. Hybrid optimization combining genetic recombination with PPO.**
> We introduce a novel hybrid optimization scheme that first recombines high scoring molecules and fine tunes the model through PPO adapted to our discrete flow models. This combination has not been explored in prior molecular generation work, and we demonstrate that it yields significant gains in sample efficiency and score on the PMO benchmark and on docking tasks.
>
> **3. Practical relevance: property optimization and docking score improvement.**
> Unlike many generative modeling papers that focus primarily on de novo generation, our work targets problems with direct practical relevance (Target property optimization/ lead optimization) and providing a new SOTA on all of them. This includes optimization against expensive oracle functions and docking based lead improvement, which are critical for real world drug discovery pipelines.
>
> **4. Fragmented SMILES design.**
> We acknowledge that the fragmentation method itself was not our intended core methodological contribution. However, our representation offers a practical advantage compared to SAFE. As illustrated figure 2, our syntax separates open cycles and attachment points in a more transparent way, which makes it easier to read and work with. This is important for practical use, since many projected users, e.g. medicinal chemists, do not have a computer science background and often find SMILES based representation difficult to interpret [1]. This is also supported by our experience of collaboration within industry. In this sense, our design lowers the barrier for working with our model, which we view as a meaningful practical improvement even if it is not the main technical novelty. Furthermore,  as denoted in the General Remarks, it allows to do linker design and scaffold morphing without any rejection sampling, something which appears to not be possible with the SAFE framework.
>
> **5. Additional minor novel contributions:**
> - Loss weighting, the benefits of which are studied in App. B.1 (Impact of Time-Weighting Loss) has not explored for discrete flows to our best knowledge
> - Integrity of PMO benchmark
> Sampling dynamics for discrete flows
> - A fragmentation pipeline with random fragment ordering for de biasing positional structure
> - A peak seeking bandit for length selection coupled to molecular reward optimization
> - A demonstration that time resolution in flows trades off quality vs diversity in a controllable way
>
> [1] Garay-Ruiz, D. & Bo, C. (2021). Human-Readable SMILES: Translating Cheminformatics to Chemistry

---

> ### Author Response · Authors · 2025-11-14
> **Weaknesses Pt. 2**
>
> > **W2. The proposed InVirtuoGen framework shows very low molecular validity in the fragment-constrained generation tasks (Table 1).**
> While we already address this in the General Remarks, we would like to provide additional clarification.
> We do not agree that validity can be described as very low when it is around 70%.
> More importantly, as we argue in our general remarks, we consider the quality metric to be far more relevant. Across all fragment-constrained tasks, our model remains competitive in terms of quality while simultaneously achieving higher diversity. In drug discovery, syntactically valid SMILES that correspond to molecules that are not synthesizable or not drug-like (or non-unique) offer little real-world value, so validity alone is not a meaningful indicator of practical performance. Quality captures the aspects that matter in practice.
> We would also like to respectfully emphasize that this is a single metric on the only benchmark among the four we evaluate on, where our method does not strictly outperform prior baselines on average. We view this not as a weakness, but as evidence that we conducted a broad and balanced evaluation across diverse tasks rather than focusing exclusively on settings where our model excels.
> Although no explicit question was posed, we hope that the clarifications above help resolve your concerns and give a clearer picture of our contributions. We would of course welcome any reconsideration this may inspire, and we remain happy to address any remaining points.

---

> ### Author Response · Authors · 2025-11-28
> **Friendly Reminder**
>
> Dear Reviewer eGk6,
>
> We hope this message finds you well. We wanted to follow up regarding our detailed response to your review, posted on November 14th.
> We have carefully addressed your concerns about novelty and validity, including:
>
> * A comprehensive clarification of our methodological contributions (refinement-based discrete flow, hybrid GA+PPO optimization, novel fragmented SMILES design)
> * Explanation of how our approach differs fundamentally from prior autoregressive and masked diffusion methods
> * Context on the fragment-constrained generation validity results and why we consider quality metrics more practically relevant
>
> We would be very grateful if you could take a moment to review our response and reconsider your assessment in light of these clarifications. If any points remain unclear or require further elaboration, we would be happy to provide additional details.
> The discussion period is drawing to a close, and your feedback would be extremely valuable.
> Thank you very much for your time and consideration.
>
> Best regards,
>
> The Authors

---

### Official Review · Reviewer_yXLN · 2025-10-31

**Soundness:** 3
**Presentation:** 3
**Contribution:** 3
**Rating:** 6
**Confidence:** 3

**Summary:**

This paper presents InVirtuoGen, a discrete flow–based molecular generator that refines fragmented SMILES strings rather than completing them autoregressively. The model starts from a uniform source and learns a refinement process that updates all tokens simultaneously. It integrates a genetic algorithm for initial candidate recombination and a PPO-based fine-tuning scheme for property optimization, achieving state-of-the-art results on PMO and competitive lead optimization performance.

**Strengths:**

1. Proposes a novel refinement-based discrete flow paradigm that departs from completion-style generation.

2. Strong empirical results across de novo, fragment-constrained, and property optimization tasks.

3. The GA + PPO hybrid optimization design is elegant and practically effective for sample-efficient molecular search.

**Weaknesses:**

1. The sampling modification (Eq. 6) lacks theoretical justification and deviates from formal flow definitions.

2. Fragment-constrained generation is not well-aligned with the model’s refinement philosophy, showing lower validity.

3. Limited analysis of training stability and efficiency vs. on-policy/off-policy baselines (e.g., GFlowNets, diffusion models).

**Questions:**

1. How does the proposed refinement training compare in sample efficiency to off-policy methods like GFlowNets?

2. Does the PPO fine-tuning use off-policy data reuse (e.g., replay) or purely on-policy updates, and how does that affect stability?

3. Could the refinement framework be extended to stochastic sampling or energy-based objectives for better exploration?

---

> ### Author Response · Authors · 2025-11-14
>
> We thank the reviewer for the thoughtful evaluation and for highlighting several of the strengths of the work. We respond to your concerns below, and we hope that the clarifications provided further strengthen the technical contribution and its positioning.
>
> > **W1: The sampling modification (Eq. 6) lacks theoretical justification and deviates from formal flow definitions.**
>
> We thank the Reviewer for pointing out this issue. To provide more intuition, we offer an empirical comparison of different sampling procedures in Appendix B.1. A key observation is that, under standard flow sampling, the number of denoising jumps becomes effectively independent of the time discretization. This raises the question of how much benefit the continuous-time formulation provides in practice, since the resulting behavior resembles that of a masked model. Prior work has already shown that continuous-time variants of masked generative models offer limited practical advantages [1], which aligns with our empirical findings.
>
> >**W2: Fragment-constrained generation is not well-aligned with the model’s refinement philosophy, showing lower validity.**
>
> While we already address this in the General Remarks, we want to give some intuition here why we believe we observe the drop in validity.
> Although both SAFE and our fragmentation pipeline use the revised BRICS algorithm, implementation differences (for example, limits on the number of fragments) yield non identical fragment sets. Because our preprocessing splits larger molecules into multiple fragments, the long, contiguous prompts used in the SAFE benchmarks rarely appear during training and are therefore out of distribution for our model. This mismatch is most apparent in the superstructure task, where the model interprets the entire prompt as editable, in contrast to masked or autoregressive models that include a bias for prompt immutability by design and are therefore naturally advantaged in this setting.
> Despite this mismatch, our model remains competitive across the fragment constrained tasks when considering metrics that better reflect practical usefulness, such as quality and diversity. We also note that GenMol tuned generation parameters for each individual task, whereas we used a single configuration across all experiments.
>
> >**W3: Limited analysis of training stability and efficiency vs. on-policy/off-policy baselines (e.g., GFlowNets, diffusion models).**
>
> This is a direction for future work, while training stability and efficiency is somewhat evaluated by the PMO benchmark [2] (which includes GFlowNets , and their improved version Genetic-guided GFlowNets). The main top AUC metric takes into account how efficient a given model is. We added a short excerpt about the stability of the optimisation routine in appendix B.3.2
>
> **Q1**: The original GFlowNets performance on PMO benchmark is given by a top-10 AUC of around 9.1, significantly lower than our performance in the non-prescreening setting of around 16.7, even when excluding the GA part of our routine, the performance with 14.3 is still significantly higher (Appendix B.3.1).
>
> **Q2**: Interestingly we found non-consistent results for the case when using pre-screening and when not using pre-screening. As we show in App B.3.1, it benefits performance when not using pre-screening. The question of stability is more difficult to answer. The behaviour of the policy varies quite strongly, even just between different random seeds as we discuss in App B.3.2. We did not find a significant drop in top AUC (which is related to stability) when using experience replay for the non prescreened version.
>
> **Q3**: We thank the reviewer for this thoughtful question. In fact, our refinement framework is already partly stochastic: each refinement step samples from the model’s categorical distribution, and the trajectory therefore explores multiple possible molecular configurations rather than following a single deterministic path. For the second part, we believe it is an interesting direction for future work, but we anticipate difficulties with introducing energy-sampling in discrete spaces.
>
> [1] Zheng, K., Chen, Y., Mao, H., Liu, M.-Y., Zhu, J., & Zhang, Q. (2024). Masked Diffusion Models are Secretly Time-Agnostic Masked Models and Exploit Inaccurate Categorical Sampling.
>
> [2] Gao, W., Fu, T., Sun, J., & Coley, C. W. (2022). Sample Efficiency Matters: A Benchmark for Practical Molecular Optimization.

---

> ### Author Response · Authors · 2025-11-28
> **Friendly Reminder**
>
> Dear Reviewer yXLN,
>
> We hope this message finds you well. We wanted to follow up regarding our detailed response to your review, posted on November 14th.
> We have carefully addressed each of your concerns and provided answers to your questions.
>
> We would be very grateful if you could take a moment to review our response and let us know if we have adequately addressed your concerns or if any points require further clarification.
> The discussion period is drawing to a close, and your feedback would be extremely valuable in helping us improve our work.
> Thank you very much for your time and consideration.
>
> Best regards,
>
> The Authors

---

### Official Review · Reviewer_BEHY · 2025-10-31

**Soundness:** 3
**Presentation:** 3
**Contribution:** 3
**Rating:** 8
**Confidence:** 3

**Summary:**

This submission investigates the problem of drug discovery, more specifically fragment-based drug discovery. The authors present a method called InVirtuoGen, which adopts the discrete flow model framework of Gat et al., 2024 and a slightly modified version of the SAFE framework of Noutahi et al., 2023 for molecule representation. The main contribution of the submission is the proposed method for Target Property Optimization on top of the Discrete Flow Model. This method combines a genetic algorithm for faster convergence with PPO-based Reinforcement Learning for gradient-based optimization. Experiments were performed on benchmarks for de novo (unconstrained) generation, fragment-constrained generation, target property optimization, and  lead optimization. Experimental results are supportive.

**Strengths:**

The submission focuses on the practically important tasks of molecule optimization instead of de novo molecule generation.

InVirtuoGen proposes a discrete flow model that transforms a uniform source distribution over all tokens into molecular data. The simultaneous generation of the whole molecule is beneficial for optimization tasks.

It enables longer generation trajectories, where the quality improves with more refinement steps.

InVirtuoGen integrates Genetic Algorithm + PPO for target-conditioned and goal-directed optimization, a novel hybrid scheme that combines the strengths of both ingredients.

Strong experimental results on high-diversity generation benchmarks, in particular for property optimization and lead optimization.

**Weaknesses:**

The number of baselines compared with experimentally is small (2 and 3, respectively), and the credibility of the experimental results would be significantly strengthened by comparing with more SOTA methods.

It remains unclear whether the experimental results for the baselines copied from the original publications are comparable with the experimental results obtained for InVirtuoGen by the authors.

Limited intuition for why validity is lower than GenMol or SAFE-GPT, even though quality metrics (valid, unique, QED ≥ 0.6) are higher.

Appears better suited for de novo or goal-directed generation, rather than fragment-constrained generation — GenMol’s masked diffusion approach respects constraints more effectively.

**Questions:**

InVirtuoGen proposes a discrete flow model that transforms a uniform source distribution over all tokens into molecular data. How is this different from starting with uniform prior, rather than a masked prior commonly done (https://arxiv.org/pdf/2402.04997)?

Most experimental results for the baselines were copied from the original publications. How did you ensure that the experimental designs are identical and the experimental results comparable?

Please perform experiments with other SOTA methods!

---

> ### Author Response · Authors · 2025-11-14
> **Addressing Weaknesses Pt.1**
>
> We thank the reviewer for the constructive and encouraging assessment, for recognizing the practical focus, the fact that our performance increases with fine sampling granularity, and the novelty of combining GA and PPO within a discrete flow framework. We try to address the raised concerns below.
> > W1. **The number of baselines compared with experimentally is small (2 and 3, respectively), and the credibility of the experimental results would be significantly strengthened by comparing with more SOTA methods.**
>
> We follow the same benchmarks as GenMol because they include the most established and widely adopted evaluation suite for practice-oriented molecular generation and optimization, to our best knowledge. This allows us to situate our method directly within the strongest existing baselines, and across these tasks our model delivers substantial and consistent gains. While there is a plethora of generative models for molecules, many of them focus on distribution learning (denovo generation), which is not the task we think is most relevant. The PMO benchmark, is in our opinion, the most practically motivated and widely adopted benchmark for goal directed molecular optimization. For this reason, it became the central focus of our evaluation. In line with GenMol, we compared against the non strictly outperformed methods rather than reproducing the full list of 30+ weaker methods reported in the original PMO study [1]
> Incorporating additional baseline models for the de novo/fragment-constrained generation is also not straightforward: to the best of our knowledge, earlier approaches do not allow to evaluate the e.g. quality/diversity tradeoff, and most operate purely on standard SMILES representations, making it unclear how to adapt them to the fragment-constrained generation task.
>
> > **W2. It remains unclear whether the experimental results for the baselines copied from the original publications are comparable with the experimental results obtained for InVirtuoGen by the authors.**
> In the following we describe how ensured a consistent evaluation on all tasks.
>
> **PMO benchmark**
> Most of those methods perform substantially worse and including them would not improve scientific soundness, while significantly increasing experimental complexity. To ensure consistency, we used the official implementation of the PMO benchmark [2] and did not interfere with oracle implementation/scoring:
>
> **De novo generation**
> All metrics used here (syntactic validity, QED, and SA score) are implemented in RDKit, ensuring full consistency across methods. Since GenMol also uses rdkit to calculate these metrics under identical assumptions, we consider the results reliably comparable.
>
> **Fragment constrained generation**
> We ensured the same fragment prompts are used, Again, all metrics (validity, uniqueness, QED > 0.6, SA<4) are computed using RDKit, making the comparison consistent.
>
> **Lead optimization**
> For lead optimization, we used the docking oracle from: https://github.com/SeulLee05/MOOD/tree/main
> Notably, this implementation was authored by the same first author as GenMol, ensuring that the oracle scoring procedure is identical.
>
> > **W3: Limited intuition for why validity is lower than GenMol or SAFE-GPT, even though quality metrics (valid, unique, QED ≥ 0.6) are higher.**
>
> First we want to clarify why we believe that the quality metric is the more informative measure. In the quality score, validity and uniqueness contribute with equal weight. This avoids degenerate cases such as a model that achieves one hundred percent validity by generating the same single molecule repeatedly, which is unusable in practice. By contrast, quality rewards models that balance validity with diverse and drug like outputs, which more accurately reflects real world utility. For these reasons, and given that additional sampling can always compensate for isolated syntactic invalidity but cannot compensate for missing diversity, we consider quality to provide a more meaningful comparison across methods. Specifically on the fragment-constrained benchmark, where our model’s uniqueness is equal to or higher than all other methods across all evaluated tasks, this highlights why we can compensate in terms of validity.
> For the other benchmarks, we would certainly like to present this behavior as a deliberate strength of our model, namely that it focuses more on generating molecules with practical relevance ( e.g. passing loose QED > 0.6 and SA < 4 cuts) rather than on producing syntactically flawless fragmented SMILES strings that may correspond to unrealistic or unsynthesizable structures. However, to be transparent, while we briefly analyze the evolution of these properties in Appendix B.1 (Comparison of Sampling Methods ), we do not yet have a conclusive explanation for the higher QED and lower SA values. At the moment, we attribute this effect to our sampling which appears to favor high QED & low SA regions (Figure 10/11 in the appendix).

---

> > ### Author Response · Authors · 2025-11-14
> > **Weaknesses Pt. 2 & Questions**
> >
> > > **W3. Appears better suited for de novo or goal-directed generation than fragment-constrained generation**
> >
> > While the validity metric may suggest that strict fragment completion is not where our model performs best, we do not present this as its primary strength, but rather its use in more practical setting. Additionally, when using quality (whilst considering the quality–diversity tradeoff) as the main metric, our model remains competitive with existing approaches across the fragment-constrained tasks. Additionally, our refinement based formulation offers a capability that masked or autoregressive models do not: it can explore the chemical neighborhood around a given fragment rather than completing it verbatim. In many real-world drug discovery workflows, the ability to generate analogs and local variations also carries their value.
> > We also emphasize that fragment-constrained generation without a target property or optimization goal has limited practical utility, and thus the performance were not our main focus. For these reasons, we do not view weaker performance on pure fragment completion as a major limitation of our approach.
> >
> > **Q1**: The answer is maybe best explained with listing 2 in the appendix of your reference (and the paragraph above it).
> > “Therefore, if x_t^d = M, then we have a probability a / (1 - t) of flipping to an unmasked state, with the probabilities for the token it unmasks to given by p_theta^{1|t}(x_1^d | x_{1:D}^t). If x_t^d is not equal to M (that is, it has already been unmasked), then it simply remains in its current unmasked state”.
> > This is similar to discrete diffusion, where once a token has been unmasked, it remains at the same state, making it more close to arbitrary order completion.
> >
> > **Q2**: We hope that we addressed this with the answer to W1
> >
> > **Q3**: We are very happy to incorporate methods that can be evaluated on the benchmarks we used, if you can provide any that we missed. But for now we are unaware what we could include that has not been already strictly outperformed by GenMol.
> > We thank the reviewer for his constructive feedback and interest for our contribution!
> >
> > [1] Gao, W., Fu, T., Sun, J., & Coley, C. W. (2022). Sample Efficiency Matters: A Benchmark for Practical Molecular Optimization.
> >
> > [2] https://github.com/wenhao-gao/mol_opt

---

> > ### Comment · Reviewer_BEHY · 2025-11-22
> >
> > Thank you very much for your thorough rebuttal!
> >
> > I appreciate your feedback regarding W2 and W3.
> >
> > Re W1
> > I agree that it does not make sense to compare against methods that were consistently outperformed by GenMol in that paper. But on the tasks where it is applicable you should compare against a GFlowNet-based method such as Rgfn:
> > Koziarski, M., Rekesh, A., Shevchuk, D., van der Sloot, A., Gaiński, P., Bengio, Y., ... & Batey, R. (2024). Rgfn: Synthesizable molecular generation using gflownets. Advances in Neural Information Processing Systems, 37, 46908-46955.

---

> > > ### Author Response · Authors · 2025-11-23
> > >
> > > We thank the reviewer for acknowledging our efforts and for pointing us to this reference. After reviewing the work, it seems that its main focus is cost efficiency and synthetic accessibility. In terms of optimization performance, meaning how much the target function improves, the underlying GFlowNet appears very similar to the one used in the PMO benchmark. For this reason, we do not expect notable performance differences, especially since one of their strongest baselines is GraphGA in terms of oracle improvement (figure 3), which we already benchmark against on PMO and lead optimization.
> > >
> > > We agree that synthesizability is important, but it was not the focus of our study, so we do not see a direct benefit from incorporating this work. Still, it remains an interesting direction for future work, and we thank the reviewer for pointing it out.

---

### Official Review · Reviewer_EgpW · 2025-11-01

**Soundness:** 2
**Presentation:** 1
**Contribution:** 3
**Rating:** 4
**Confidence:** 4

**Summary:**

This paper introduces InVirtuoGen, a novel generative model for fragment-based drug discovery. The core contribution is a paradigm shift from "auto-regressive" used by SAFE-GPT or "completion" used by GenMol to "refinement." Authors investigate the model performance on four drug discovery tasks: de-novo generation, fragment-constrained generation, property optimization, and lead optimization. The method achieves a superior quality-diversity pareto frontier in de novo generation and sets a new state-of-the-art (SOTA) on the Practical Molecular Optimization (PMO) benchmark under both 260k and 10k oracle calls.

However, there are some concerns, and I would like to increase the score depending on the response.

---
**Usage of LLM:** I wrote the entire review myself and only used the LLM to correct the grammar and improve readability.

**Strengths:**

- **Novel & Versatile Generation Paradigm**: Authors proposed a novel paradigm for molecular generation. In particular, it is different previous auto-regressive approach or unmasking approach.

- **Commitment to Fair Benchmarking**: A major strength is the paper's transparent and fair evaluation. For the PMO benchmark, the authors correctly identify that prior SOTA models (GenMol, f-RAG) used a 250,000-call pre-screening step without fully accounting for it. This paper reports results in both settings (with and without pre-screening), providing a much fairer comparison against all baselines (e.g., Genetic GFN) and restoring integrity to the benchmark.

- **Transparency in Reporting Failures**: The authors are commendably transparent about the model's full capabilities, rather than cherry-picking only successful tasks. They include a detailed analysis of fragment-constrained generation, a task where the model performs poorly and its validity drops significantly. This transparency provides valuable insights into the model's "philosophy" and its operational boundaries.

**Weaknesses:**

**1. Lack of Conditional Generative Capabilities**

The paper successfully employs a PPO-based fine-tuning strategy for property optimization. However, unlike baseline model SAFE-GPT, authors do not investigate conditional generative capabilities proposed by Scaffold-GGM[1] and BBAR[2]. The ability to control key molecular properties directly (e.g., "generate a molecule with LogP = 2") is an important task. Demonstrating this capability would have significantly strengthened the paper's claim of being a "versatile generative foundation."

**2. Misleading Paper Structure**

The paper's structure is misleading. The fragment-constrained generation task is presented early (Section 3.2, Table 1), suggesting it is a core capability. However, this task yields extremely low molecular validity (e.g., 28.6% for superstructure design) and, as the authors rightly admit, fundamentally "conflicts with the refinement philosophy." This prominent placement of a failed task creates significant doubt about the model's reliability. I suggest moving this section to the last section of manuscript or appendix in a camera-ready version.

**3. Lack of validity metrics**

Related to W2, authors did not report independent validity metrics for its other, more successful tasks (de novo generation and property optimization). In the de novo task (Sec 3.1), validity is conflated with other metrics inside the "Quality" score, making it impossible to assess the base generative success rate. In the optimization tasks (Sec 3.3, 3.4), only the final scores of successful molecules are reported, giving no insight into the efficiency of the process (i.e., how many invalid molecules were generated and discarded to achieve those scores). This omission is significant, as it prevents a full assessment of the model's generative reliability in its primary use cases.

---
**References:**
1. Lim, Jaechang, et al. "Scaffold-based molecular design with a graph generative model." Chemical science 11.4 (2020): 1153-1164.
2. Seo, Seonghwan, Jaechang Lim, and Woo Youn Kim. "Molecular generative model via retrosynthetically prepared chemical building block assembly." Advanced Science 10.8 (2023): 2206674.

**Questions:**

- Line 153: How is `c1ccccc1` tokenized? `c1` or `c`&`1`?
- Line 283: As I know, GenMol cut an arbitrary single bond in molecules to create offspring. Since this cutting can be conducted with a reaction template, it does not restrict the chemical space.

---

> ### Author Response · Authors · 2025-11-14
> **Adressing Weaknesses**
>
> We sincerely thank the reviewer for the constructive and thorough feedback, and for acknowledging both the novelty of the refinement paradigm and our effort to restore integrity of the PMO benchmark. We address each concern in detail below.
> >  **W1. Lack of Conditional Generative Capabilities**
>
> We thank the reviewer for highlighting this point, and we agree that conditional control ( e.g., generating molecules with a target LogP or high QED) is important. We initially experimented with conditional training, but ultimately chose not to include it in this version of the work because it was unclear how to integrate it without compromising the fairness of our benchmark evaluations. For instance, one of the PMO tasks explicitly maximizes QED. If our model were pretrained with QED as a conditioning feature, this would implicitly give it access to billions of effective oracle evaluations, making comparisons against existing baselines no longer meaningful.
> However, we now include a dedicated evaluation in Appendix B.5, where we demonstrate that the model reliably follows QED conditioning and clarify how this differs from target-property optimization. Conditioning is implemented via a lightweight conditioning layer whose output is added to the timestep embedding. The results were obtained by briefly fine-tuning the smaller checkpoint, following the standard pretraining setup but providing the condition as an additional model input. We expect that conditioning from the start of training, particularly when combined with recent advances in guidance for discrete flow and diffusion models [1], would further improve adherence to target properties. We thank the reviewer for the references, which we have added to the revised manuscript.
>
> >  **W2. Misleading Paper Structure**
>
> Although both SAFE and our fragmentation pipeline rely on the revised BRICS algorithm, implementation details (such as the maximum number of allowed fragments) produce non-identical fragment sets. Because our preprocessing typically splits larger structures into multiple fragments, many of the long, contiguous prompts used in the SAFE benchmarks never appear in that form during training, making them out of distribution for our model.
> This mismatch is most visible in the superstructure generation task, where the input is a single long SMILES string. Since InVirtuoGen is trained exclusively on fragmented SMILES, the refinement model interprets the entire prompt as editable and may refine it, rather than treating it as fixed. Masked or autoregressive models, by contrast, enforce prompt immutability by design, which naturally favors them in such tasks.
> While our quality metric on this specific task (28.7%) is lower than GenMol’s (34.8%), the generated samples are still valid in 74.9% of the cases. More importantly, when considering all fragment-constrained tasks, our model remains strongly competitive in terms of both quality and diversity - which in our opinion are more meaningful. We also want to point out that GenMol tuned the parameters for every task, something we abstained from doing.
> Following the Reviewer’s suggestion, we have moved the fragment-constrained generation experiments to the appendix and provide only a short summary in the main text. We also address the observed differences in the revised manuscript.
>
> > **W3. Validity Metrics and Efficiency of Optimization**
>
> While we stand by our point that the quality metric already incorporates validity and is a more practice oriented indicator, we agree that reporting validity in isolation provides useful insight into model behavior. We have added these results to Appendix B.1 for the de novo setting.
> For target property optimization, the bottleneck is the oracle, not the generator: generating additional molecules is effectively free compared to e.g. running docking, free energy estimations, or even binding assays. Thus, optimization efficiency is governed by oracle calls rather than generative validity. Nevertheless, we track validity throughout fine-tuning to detect policy collapse internally and to terminate the PPO+GA routine. We now include validity curves as a function of oracle calls in Appendix B.3.2 for the (non-cherry picked) ranolazine experiment. These curves show that two runs that differ only in the random seed can exhibit noticeably different validity behavior. This seed sensitivity highlights a degree of instability in the current optimization procedure, which we plan to address in future work, for instance by incorporating stabilizing techniques such as BAPO [2].
>
> ### References:
> [1] Nisonoff et al., 2025. Unlocking Guidance for Discrete State-Space Diffusion and Flow Models.
>
> [2] Xi et al., 2025. BAPO: Stabilizing Off-Policy Reinforcement Learning for LLMs via Balanced Policy Optimization with Adaptive Clipping.

---

> ### Author Response · Authors · 2025-11-14
> **Adressing Questions**
>
> Questions:
>
> **Q1**: `c1` is encoded as two separate tokens, we chose a more atomistic vocabulary, since the sequence length (around 70 on average) of the samples is not a core limiting factor.
>
> **Q2**: We would like to thank the Reviewer for pointing this out, our phrasing comes across much stronger than intended. As correctly stated by the Reviewer GenMol implements offspring generation by cutting arbitrary single bonds and reconnecting fragments using a generic coupling SMARTS `([*:1]-[1*].[1*]-[*:2] >> [*:1]-[*:2])`, rather than a narrow library of specific reaction templates. Our intention was not to imply that GenMol severely restricts chemical space, but their template prescribes both how fragments are joined (always via a single bond) and where they must be joined (at dummy attachment atoms `[1*]` that must be present in the fragments). As a consequence, the attachment geometry is fixed by construction.
>
> By contrast, our refinement based approach operates directly on the tokenized representation of the molecule and does not require predefined attachment points or reaction definitions (i.e. allowing the connection of two fragments, even if no attachment point is defined). This allows the model to not only connect fragments but also adjust and reorganize the surrounding chemical environment during refinement. In practical terms, this means that our model can explore the neighborhood of a fragments rather than being forced to connect it at a single predefined location.
>
> We hope that we addressed the Reviewer’s concerns, if anything remains unclear, we would kindly ask him/her to follow up!

---

> ### Comment · Reviewer_EgpW · 2025-11-17
> **Thank you for your response**
>
> Dear authors,
>
> Thank you for detailed response. I'll increase my score.
>
> Response to W1: I simply suggested that performing a task conduced by SAFE-GPT would be beneficial, and a comparison study is not necessary. The results are quite impressive. Thank you for conducting this experiment.
>
> Response to W2: Rather than intending to point out the shortcomings of InvirtuoGen compared to other models, I was concerned that this placement might prevent the paper's contribution from being properly recognized.
>
> Response to W3: Thank you for providing detailed analysis!

---

> > ### Author Response · Authors · 2025-11-17
> >
> > Thank you very much for taking the time to reassess our submission and for increasing your score. We sincerely appreciate it.
> >
> > Regarding the question of reproducing SAFE-GPT results, we would like to clarify that we are not aware of any officially released reproduction scripts or evaluation code for the fragment-constrained generation benchmarks used in the SAFE-GPT paper. The only public repository we could locate is:
> >
> > https://github.com/datamol-io/safe
> >
> > which contains the SAFE tokenizer and the fragments used in the fragment-constrained generation, but does not include the code, instructions, or evaluation scripts needed to reproduce the fragment-constrained generation results reported in the paper. If there exists an additional reproduction resource that we have overlooked, we would be grateful for a pointer and would be happy to rerun our comparisons.
> >
> > We thank the reviewer again for the thoughtful feedback and helping further refine our work!

---

> ### Comment · Reviewer_EgpW · 2025-11-17
>
> You don't have to reproduce or compare with SAFE-GPT. Current conditioning results are already great.
>
> Regards,
> Reviewer

---

### Author Response · Authors · 2025-11-14
**General Remarks**

# General Remarks

We thank all reviewers for their careful reading, constructive feedback, and positive remarks on the empirical strength and practical relevance of InVirtuoGen. Across the reviews, we see broad agreement that the refinement based discrete flow formulation, the GA plus PPO optimization scheme, and the performance on PMO and lead optimization are promising.

## Practical Focus on Quality and Diversity, not Raw Validity

Several reviewers noted lower validity in some tasks. While we agree that validity is an important metric, we want to emphasize that in practical applications the central question is whether the generated molecules are useful, that is, synthesizable, drug-like, and diverse. Our quality metric combines validity and uniqueness in a way that directly reflects this. It excludes degenerate edge cases where a model generates the same valid molecule repeatedly, or where only a single valid structure appears among many repeats. We also note that a moderate drop in validity from 95 percent to 75 percent requires only about 1.3 times more samples to recover the same number of valid molecules, whereas low diversity cannot be compensated in this way.

## Fragment-Constrained Generation

Since multiple reviewers raised concerns about our performance on fragment-constrained tasks, we thoroughly investigated observed differences. In the following, we want to highlight specific issues with baseline comparability that we found.

### GenMol

While Fig. 2 of the GenMol paper suggests that the only difference between de novo generation and fragment-constrained generation is the use of a fixed prompt, the public implementation reveals that GenMol performs fragment constrained linker design and scaffold morphing through multi step rejection sampling, not direct conditional completion. The function starting at:

https://github.com/NVIDIA-Digital-Bio/genmol/blob/f384e1e05c2f810302886d637ea5f0b558b9be3a/src/genmol/sampler.py#L130C1-L150C23

splits each fragment into a prefix and a suffix, generates both sides independently, and then reconstructs molecules through mix_sequences and filter_by_substructure, silently discarding invalid, unparsable or non matching generations. Since only the surviving reconstructions are returned, the reported validity excludes all failed generations and is therefore not comparable to raw validity numbers reported for other models.

In a recent update in the repository README, the authors acknowledge this and report results without the rejection based filtering. Under this more fair setting, their validity and quality on linker design and scaffold morphing drop to **very low** levels. To be completely open, they also introduce a new SAFE syntax to mitigate this, but the updated formalism is not accompanied by a publication and the timeline for evaluation is too short. Moreover, GenMol V2 shows a substantial drop in performance on the PMO benchmark. Taken together, these observations strengthen our results because they suggest that the original GenMol numbers in the paper cannot be used as a reliable point of comparison for fragment constrained generation tasks.

### SAFE-GPT

SAFE-GPT also employs constrained decoding tailored to fragment constrained tasks. Appendix A.4 describes constrained beam search to enforce fragment presence and connectivity constrained generation for scaffold morphing and linker design. Since no unconstrained results are reported, comparisons against GenMol should be interpreted with care. These constraint mechanisms strongly affect validity and should be made explicit when comparing models.

We include the updated table below, which reflects a more consistent comparison across models. We hope this resolves the concerns raised about our performance on this specific task. We include the updated results in the revised paper.

---

> ### Author Response · Authors · 2025-11-16
>
> | Task                   | Method        | Diversity | Quality | Uniqueness | Validity |
> |------------------------|---------------|-----------|---------|------------|----------|
> | **Motif Extension**    | SAFE-GPT      | 0.56      | 18.60   | 66.80      | **96.10** |
> |                        | GenMol        | **0.62**  | 30.10   | 77.50      | 82.90     |
> |                        | InVirtuoGen   | **0.62**  | **39.27** | **96.83** | 68.97     |
> | **Linker Design**      | SAFE-GPT      | **0.55**  | **21.70** | 82.50    | **76.60** |
> |                        | GenMol        | 0.53      | 4.30    | **97.80**  | 16.70     |
> |                        | InVirtuoGen   | 0.52      | **22.33** | 84.76    | 60.37     |
> | **Scaffold Morphing**  | SAFE-GPT      | 0.51      | 16.70   | 70.40      | 58.90     |
> |                        | GenMol        | **0.53**  | 4.30    | **97.80**  | 16.70     |
> |                        | InVirtuoGen   | 0.52      | **22.33** | 84.76    | **60.37** |
> | **Superstructure Design** | SAFE-GPT   | 0.57      | 14.30   | 83.00      | 95.70     |
> |                        | GenMol        | 0.60      | **34.80** | 83.60   | **97.50** |
> |                        | InVirtuoGen   | **0.73**  | 27.43   | **99.41**  | 75.70     |
> | **Scaffold Decoration** | SAFE-GPT     | 0.57      | 10.00   | 74.70      | **97.70** |
> |                        | GenMol        | **0.59**  | 31.80   | 82.70      | 96.60     |
> |                        | InVirtuoGen   | 0.56      | **36.37** | **88.58** | 90.70     |
> | **Average**            | SAFE-GPT      | 0.55      | 16.26   | 75.48      | **85.00** |
> |                        | GenMol        | 0.57      | 21.06   | 87.88      | 62.08     |
> |                        | InVirtuoGen   | **0.59**  | **29.55** | **90.87** | 71.22     |

---

### Meta-Review · Area_Chair_bFp5 · 2026-01-07

**Summary:**

1. Multiple reviewers raised concerns about the paper's structure and clarity. Reviewer EgpW found the placement of fragment-constrained generation misleading since the model performs poorly on this task, suggesting it conflicts with the refinement philosophy. The lack of clearly separated method sections made it difficult for some reviewers to parse the core contributions. Reviewer eGk6 was particularly critical, viewing the methodological novelty as weak and noting the method section was essentially missing from the main paper.

2. The lower validity scores compared to baselines like GenMol and SAFE-GPT troubled several reviewers, even though the authors argue convincingly that quality metrics combining validity with uniqueness and drug-likeness matter more practically. Reviewer EgpW noted validity wasn't reported independently for successful tasks like de novo generation and property optimization, making it hard to assess generative reliability. The authors addressed this by adding validity results to appendices, but the initial omission raised questions about transparency.

3. Fragment-constrained generation performance emerged as the most significant weakness across reviews. The model shows substantially lower validity on fragment-constrained tasks, which the authors acknowledge stems from their refinement approach treating prompts as editable rather than fixed. While they provide reasonable explanations about training data distribution mismatches and argue this task isn't practically critical, the weak performance on a prominently featured benchmark undermines confidence in the model's versatility claims.

4. Reviewer BEHY questioned whether copied baseline results from original publications were truly comparable to the authors' experiments. The authors provided detailed explanations about using identical evaluation codebases and metrics, which seems adequate. However, the revelation that GenMol's fragment-constrained results used hidden rejection sampling that inflated their reported validity raises broader concerns about benchmark integrity, though this actually strengthens the authors' position.

5. Reviewer yXLN pointed out that the modified sampling procedure lacks theoretical grounding and deviates from formal flow definitions. The authors provided empirical comparisons but acknowledged this remains somewhat unprincipled. This theoretical gap, combined with limited analysis of training stability compared to off-policy methods like GFlowNets, suggests the approach may be more heuristic than principled.

6. The paper tackles multiple distinct problems, de novo generation, fragment-constrained generation, property optimization, and lead optimization, which creates both breadth and dilution of focus. Strong performance on PMO benchmark and lead optimization is impressive, but the fragment-constrained weaknesses and questions about conditional generation capabilities suggest the "versatile foundation" framing may overreach. Reviewer EgpW's suggestion to add conditional generation experiments was constructive and the authors complied, but this highlights gaps in the initial submission.

**Reviewer Concerns:**

Concerns Adequately Addressed by Rebuttal
1. Reviewer EgpW's concern about lack of conditional generation capabilities was resolved convincingly. The authors added experiments in Appendix B.5 demonstrating QED conditioning works reliably, explained why they avoided it initially to maintain benchmark fairness, and incorporated the suggested references. The reviewer increased their score and explicitly stated the conditioning results were impressive.
2. The baseline comparability question from Reviewer BEHY was handled well. The authors provided detailed documentation of using identical evaluation frameworks, the official PMO benchmark implementation, and RDKit metrics across all methods. More importantly, they revealed that GenMol's fragment-constrained results used undisclosed rejection sampling, which actually undermines that baseline rather than their own work. This strengthens their position considerably.
3. Reviewer EgpW's structural concern about misleading paper organization was addressed by moving fragment-constrained generation to the appendix in the revision. While this doesn't fix the underlying performance issue, it resolves the presentation problem of prominently featuring a task where the model struggles.
4. The validity metrics omission raised by multiple reviewers was corrected by adding validity results to appendices for de novo generation and including validity curves for property optimization. The authors also provided clearer argumentation for why quality metrics matter more practically, which is defensible even if not universally persuasive.
5. Questions about training efficiency versus GFlowNets from Reviewer yXLN were answered with concrete numbers showing their top-10 AUC of 16.7 substantially exceeds GFlowNets' 9.1 on PMO, demonstrating superior sample efficiency empirically if not theoretically.

Outstanding Concerns Not Fully Resolved
1. The fundamental weakness on fragment-constrained generation remains unresolved despite extensive explanation. While the authors provide reasonable justifications about training distribution mismatches and argue this task isn't practically critical, the performance gap is substantial, validity around 60-70% versus 80-95% for baselines. Their argument that refinement philosophy conflicts with prompt immutability essentially concedes this is an architectural limitation rather than something addressable through better training or sampling.
2. Reviewer yXLN's concern about lack of theoretical justification for the modified sampling procedure was only partially addressed. The authors provided empirical comparisons in Appendix B.1 and noted that standard flow sampling resembles masked models, but they didn't provide rigorous theoretical grounding. The admission that the approach may be more heuristic than principled remains a legitimate concern for a venue emphasizing foundational contributions.
3. Reviewer eGk6's core criticism about weak novelty was addressed with extensive enumeration of contributions, but whether this is convincing depends on perspective. The discrete flow adaptation, hybrid GA plus PPO scheme, and practical relevance are genuine contributions, but Reviewer eGk6 viewed the fragmented SMILES as minor and the overall combination as incremental. The rebuttal doesn't fundamentally change the nature of the contributions, it just articulates them more clearly. A reviewer predisposed to value empirical results would be satisfied, one emphasizing methodological novelty might not be.
4. The training stability analysis requested by Reviewer yXLN was only superficially addressed. The authors mention stability is evaluated through PMO benchmark and add brief comments in Appendix B.3.2 noting that different random seeds produce noticeably different validity behavior, which they acknowledge as instability requiring future work. This actually confirms rather than resolves the concern about insufficient analysis of training dynamics.
5. Reviewer BEHY's request for additional baselines beyond the limited 2-3 comparisons per task wasn't fully satisfied. The authors argue GenMol already outperforms most alternatives and adding weaker baselines wouldn't improve scientific soundness, which is reasonable but doesn't address whether other strong recent methods exist. The suggestion to compare against RGFlowNet was deflected by arguing it focuses on synthesizability rather than optimization performance, but this seems somewhat evasive given that work also reports optimization results.
6. The question from Reviewer yXLN about whether PPO uses off-policy replay received a mixed answer. The authors note inconsistent results between prescreening and non-prescreening settings, mention policy behavior varies strongly between random seeds, and state they didn't find significant top-10 AUC drops with experience replay. This suggests they don't fully understand the stability dynamics, which undermines confidence that the optimization procedure is robust and well-characterized.

The overarching issue is that the paper claims to provide a versatile generative foundation for drug discovery spanning early hit finding to lead optimization, but demonstrates clear weakness on fragment-constrained generation, a task that falls squarely within this claimed scope. The authors' strategy of downplaying this task's practical importance and emphasizing quality over validity is defensible but doesn't eliminate the tension between their broad claims and demonstrated limitations. A more modest framing focused specifically on property optimization and lead refinement would be more aligned with the empirical evidence.

**Reviewer Scores:**

Reviewer EgpW (Initial Score: 4 → Actual Final Score: Not explicitly stated but implied increase)
This reviewer explicitly stated "I'll increase my score" after the rebuttal and clarified their concerns were about presentation rather than fundamental flaws. Based on their positive tone in follow-up comments ("The results are quite impressive. Thank you for conducting this experiment" regarding conditioning, and "You don't have to reproduce or compare with SAFE-GPT. Current conditioning results are already great"), I estimate they would move to a 6.

Reviewer BEHY (Initial Score: 8)
This reviewer was already positive and their follow-up engagement was constructive rather than critical. They thanked the authors for the "thorough rebuttal" and their only substantive request was to compare against RGFlowNet, which the authors reasonably explained focuses on different aspects. They would likely remain at 8.

Reviewer yXLN (Initial Score: 6 → Likely Final Score: 6)
This reviewer never responded to the authors' rebuttal despite a friendly reminder, which is concerning. Their initial score of 6 reflected "marginally above the acceptance threshold" with legitimate technical concerns about theoretical justification, training stability analysis, and sampling modifications.

Reviewer eGk6 (Initial Score: 2 → Likely remain same or best case scenario final score: 4)
This is the most difficult to predict because they also didn't respond despite a friendly reminder. Their initial harsh assessment (2 score, "poor" on all dimensions) focused on weak novelty and missing method sections. The rebuttal provided extensive clarification of contributions and reorganized the paper structure, which addresses the presentation concern. However, their fundamental view that combining discrete flows with fragmented SMILES constitutes incremental rather than significant contribution is philosophical and unlikely to change dramatically.

---

### Decision · Program_Chairs · 2026-01-26

Accept (Poster)